# Pathologic and Therapeutic Schwann Cells

**DOI:** 10.3390/cells14171336

**Published:** 2025-08-28

**Authors:** Michael R. Shurin, Sarah E. Wheeler, Hua Zhong, Yan Zhou

**Affiliations:** 1Division of Clinical Immunopathology, Department of Pathology, University of Pittsburgh and University of Pittsburgh Medical Center, Pittsburgh, PA 15213, USA; wheelerse3@upmc.edu; 2Department of Respiratory and Critical Care Medicine, Shanghai Chest Hospital, Shanghai Jiao Tong University School of Medicine, Shanghai 200025, China; eddiedong8@163.com (H.Z.); yzhou716@163.com (Y.Z.)

**Keywords:** Schwann cells, neuropathy, cellular therapy, neurotrauma, nerve repair, neuropathic pain, neuroinflammation

## Abstract

Schwann cells (SCs) are the primary glial cells of the Peripheral Nervous System (PNS), which insulate and provide protection and nutrients to the axons. Technological and experimental advances in neuroscience, focusing on the biology of SCs, their interactions with other cells, and their role in the pathogenesis of various diseases, have paved the way for exploring new treatment strategies that aim to harness the direct protective or causative properties of SCs in neurological disorders. SCs express cytokines, chemokines, neurotrophic growth factors, matrix metalloproteinases, extracellular matrix proteins, and extracellular vesicles, which promote the inherent potential of the injured neurons to survive and accelerate axonal elongation. The ability of SCs to support the development and functioning of neurons is lost in certain hereditary, autoimmune, metabolic, traumatic, and toxic conditions, suggesting their role in specific neurological diseases. Thus, targeting, modifying, and replacing SC strategies, as well as utilizing SC-derived factors and exosomes, have been considered novel therapeutic opportunities for neuropathological conditions. Preclinical and clinical data have demonstrated that SCs and SC-derived factors can serve as viable cell therapy for reconstructing the local tissue microenvironment and promoting nerve anatomical and functional recovery in both peripheral and central nerve injury repair, as well as in peripheral neuropathies. However, despite the promising successes of genetic engineering of SCs, which are now in preclinical and clinical trials, improving tactics to obtain ‘repair’ SCs and their products from different sources is the key goal for future clinical success. Finally, further development of innovative therapeutic approaches to target and modify SC survival and function in vivo is also urgently needed.

## 1. Introduction

Schwann cells (SCs), the primary glial cells in the peripheral nervous system (PNS), exhibit two key phenotypes: myelinating and non-myelinating. They originate from embryonic Schwann cell precursors (SCPs) derived from the neural crest, which travel and proliferate along the PNS axonal tracts. SCPs differentiate into immature SCs, which then become pro-myelinating SCs that are eventually converted into either myelinating or non-myelinating SCs [1]. During embryogenesis, SCs safeguard neuronal longevity and participate in axonal pathfinding. Simultaneously, they orchestrate the architectural setup of the developing nerves, including the blood vessels and all layers of the peripheral nerves [2]. Later, SCs enable the transmission of neural impulses, deliver nutrients and shelter for neurons, guide axons in nerve repair, and control inflammatory and immune responses in different tissues and organs by enwrapping both myelinated and non-myelinated axons and cross-talking with cells in the local microenvironment [3]. While most nerve fibers are unmyelinated in the PNS, all nerve fibers of the PNS are enveloped by SCs: axons are either bound by a single-layered SC cell membrane to make non-myelinated nerve fibers or enfolded by several coatings of SC membranes to form myelinated nerve fibers [4,5]. Although non-myelinating SCs do not make myelin, they participate in essential functions of the PNS, including the maintenance of axonal metabolism and control of neuropathic pain. Non-myelinating SCs also possess the potential for myelination, as they can transit to repair SCs and initiate the repair process during nerve injury. Non-myelin-forming SCs may likely act as the “first responders” to traumatic insults or illnesses in their local environment [6].

SCs participate in the pathogenesis of various dysmetabolic, traumatic, and hereditary conditions and diseases of the PNS (Figure 1). SCs may initiate pain and modulate pain sensitivity in both physiological and pathological conditions [7,8]. The role of specific receptors, ion channels, and bioactive molecules in mediating neuropathic pain by SCs has been well described [9]. Notably, a specialized mesh-like network of cutaneous SCs that conveys noxious thermal and mechanical sensitivity has been recently reported [10]. These sensory SCs could transmit nociceptive information to the closely associated unmyelinated nociceptive nerves. In addition, SCs, specifically perisynaptic non-myelinating SCs, participate in the formation and function of the neuromuscular junction, are involved in polyneuronal innervation and synapse elimination, and can alter the synaptic transmission [11].

SCs are also known to function as immunomodulating cells because they can process and present antigens, produce cytokines and chemokines, and express pattern recognition receptors, such as Toll-like receptors (TLRs) and the nucleotide-binding and oligomerization domain (NOD)-like receptors (NLRs) [12,13]. SCs were identified in the thymus and spleen, where they directly interact with lymphocytes and dendritic cells under both normal and pathological conditions [14,15]. For instance, the interaction between SCs and macrophages is crucial for orchestrating tissue regeneration after peripheral nerve injury [16]. Human repair SCs are phagocytic; express MHC-II, CD40, CD80, CD86, B7H3, and PD-L1; release chemoattractants, matrix remodeling proteins, and pro- and anti-inflammatory cytokines; and may serve as regulators of T-cell immunity in nerve regeneration [17]. Interestingly, in the bone marrow, non-myelinating SCs may be an essential component of the bone marrow niche responsible, at least in part, for maintaining hematopoietic stem cell (HSC) hibernation by regulating activation of latent transforming growth factor beta (TGF-β), as autonomic nerve denervation has been reported to reduce the number of TGF-β-producing SCs, leading to rapid loss of HSCs from the bone marrow [18].

SCs play a key role in PNS nerve regeneration after trauma or injury by controlling nerve fiber regeneration, myelination, and axonal guidance [19]. They also participate in the formation of a “nerve bridge” between the proximal and distal stumps of a damaged nerve, seen as a 3D structure of fibroblasts, perineurial and inflammatory cells, and a matrix [20]. The metabolic switch in SCs and metabolic coupling between SCs and axons deliver pyruvate and lactate to damaged neurons to prevent axonal degeneration [21]. These complex steps of Wallerian degeneration, or programmed axon degeneration, and axon regeneration are orchestrated by several SC-targeting factors, which polarize SCs into a reparative phenotype and control SC interaction with neurons, immune cells, fibroblasts, and endothelial cells [20,22]. Dedifferentiation, activation, proliferation, and migration of SCs upon injury, recognized as SC plasticity [2], result in the recruitment of immune cells, breakdown of myelin, elimination of debris, removal of dead cells, promotion of axonal recovery, and subsequent myelination of regenerated nerve fibers [23,24]. However, the neural repair ability of SCs may be reduced with age [25].

Furthermore, the incredible plasticity of repair SCs is not limited to their role in PNS nerve regeneration. Contemporary data have revealed that SCs possess a broad capacity to promote the restoration and regeneration of various adult tissues, in addition to PNS axons [26]. For example, the use of a mouse model of excisional cutaneous wounding helps reveal that SCs may also induce adult skin wound healing: genetic excision of SCs resulted in suspended wound contraction and closure, inhibited myofibroblast formation, and weakened skin re-epithelialization after damage [27]. These data may also highlight the importance of better understanding the connections between melanocytes and SCs in development and transformation [28]. Furthermore, new data have demonstrated the dedifferentiation of SCs after neighboring alveolar bone injury and their contribution to bone regeneration [29]. At the molecular level, the injury-induced activation of SCs is associated with the expression of various factors that regulate epithelial–mesenchymal transitions (EMT), stemness, and differentiation states linked to and contributing to injury-induced tissue remodeling [30].

Understanding the role of SCs in the functioning of the PNS and their interactions with various cell types in different tissues and organs, as well as their contribution to the pathogenic mechanisms of neurodegenerative diseases, peripheral neuropathy, neuroinflammation, certain infections, pain, and cancer, is not surprising. In the PNS, for instance, various insults, such as mutations, lesions, autoimmune reactivity, and infections, can damage the myelination process, alter the functions of SCs, and ultimately lead to neurodegeneration (Figure 1). Distraction of SCs is seen in different peripheral demyelinating disorders, such as Charcot–Marie–Tooth (CMT) disease, where myelinating SC dysfunction is attributed to the mutations of genes encoding the myelin proteins and leads to segmental demyelination (the loss of myelin sheaths after their proper development), dysmyelination (improper development of myelin sheaths), muscle dystrophy, and sensory loss [31]. Another example is hyperglycemia-induced SC damage in diabetic neuropathy that results in the loss of trophic support from SCs and damage to both sensory and motor nerves [32]. Interestingly, even though demyelinating diseases have different etiologies, such as inherited, autoimmune, inflammatory, or toxic flaws in myelin development, the concept of the “demyelinating SCs” as a specific cell phenotype required for myelin sheath clearance has been recently announced [33,34].

Thus, a thorough understanding of SC behavior in nerve regeneration, wound healing, neuropathies, and associated pathophysiological conditions may provide a practical tool to accelerate and direct the regenerative process and create the grounds for the development of new glia-centric treatment approaches to counteract axonal loss, pain, neuroinflammation, and potentially tumorigenesis [35]. Glial cells are approximately 10 times more numerous than neurons [36]. Remarkably, several reports suggest that SCs and SC-like cells are among the most promising cell subsets for cell-based therapies [30,37,38,39,40].

## 2. Schwann Cells in CNS Pathology

The nervous system’s hierarchical organization enables the control of peripheral functions and facilitates a wide range of interactions between the central nervous system (CNS) and the PNS. The confirmed presence of SCs in the CNS in various pathophysiological conditions, along with their reported beneficial effects on central remyelination, opens exciting opportunities for developing rational therapeutic approaches that utilize SCs in CNS repair.

### 2.1. Schwann Cells for Treating Central Neurotrauma

Traumatic brain injury and spinal cord injury are devastating conditions that lead to profound disability and many related complications. Due to the limited regenerative capacity of the CNS, there is currently no effective cure for these disorders. Developed treatments, such as cell transplantation, could be an encouraging opportunity for treating affected patients, based on the proven role of resident SCs in the remyelination of the spinal cord after injury. In 1981, Duncan et al. demonstrated in a focal demyelination mouse model that SCs can myelinate spinal cord axons [41]. In 1988, Berry et al. reported that SCs participated in axon regeneration within peripheral nerve grafts transplanted into the CNS [42]. Even though SCs do not occur endogenously in the CNS, many new data suggest that non-myelinating and myelinating SCs can be engaged from the PNS to help myelin repair after CNS injury [15,43,44,45]. Using the model of demyelination and paraplegia in rats, a spontaneous entry of SCs into the spinal cord, along with axonal remyelination, concomitant functional recovery from paraplegia, and SC myelin replacement by oligodendrocyte myelin, has been demonstrated [46]. Furthermore, endogenous SCs can be encouraged to migrate by minimal insults within the spinal cord and integrate with astrocytes under certain conditions [47]. Interestingly, new data demonstrate that SC migration to demyelinated CNS lesions may be guided by blood vessels [48]. Oligodendrocyte precursor cells (OPCs) in the CNS may undertake fate-switching to differentiate into remyelinating SCs after demyelination and nerve damage [49,50,51]. These and other data revealed that resident SCs could reverse a critical neurological deficit caused by CNS demyelination, strongly supporting the capacity of SCs to contribute to central nerve injury repair [45]. Intriguingly, it has been postulated that SC-remyelinated axons may be more resistant to demyelinating diseases of the CNS [51].

The benefits of transplanting cultured SCs for spinal cord injury therapy have been thoroughly investigated in experimental animals since the early 1990s. Numerous preclinical and clinical studies provide evidence that the transplantation of SCs is a feasible therapeutic approach for spinal cord repair (Figure 1 and Figure 2). The utility of SC transplantation over the last few decades has primarily focused on paradigms related to spinal cord injury and, to some extent, traumatic injury of the optic nerves and peripheral nerves [52,53,54]. For instance, it has been demonstrated that SC transplantation promotes axon regeneration from retinal ganglion cells and peripheral-type myelination in the injured optic nerve or spinal cord in a rat model [55,56]. Many studies have successfully utilized SCs for spinal cord repair and advancing functional rehabilitation in animal models of spinal cord injury [57,58,59,60,61].

It was estimated that more than 70% of engrafted SCs were myelinating in the chronically injured spinal cord. In addition to myelinating axons, grafted SCs engage resident SCs to myelinate the injured axons [61]. Furthermore, grafted SC participation in remyelination and promotion of nerve restoration has been demonstrated in a rat cerebral hemorrhage model [62]. The administration of SCs into the injured spinal cord has been reported to reduce tissue loss, support the survival of damaged neurons, encourage axonal regeneration, and accelerate myelination of axons, thereby improving sensorimotor function [38]. For instance, one report revealed a two-fold rise in the number of protected neurons within the lesion after SC transplantation [63]. The ability of SCs to restrain the pro-inflammatory microglia and macrophage phenotypes after intraspinal transplantation has also been reported [30,64], which may contribute significantly to the neuroprotective effects of SCs within the subacute spinal cord injury setting. Administration of SCs may also reduce the activation of inflammasome complexes [65]. Finally, in a xenotransplant model, human SCs have been shown to persist within the contused nude rat spinal cord for six months after transplantation, exhibiting no evidence of tumorigenicity, showing a low proliferation rate, and displaying a controlled biodistribution to the lesion [66].

Results of several clinical trials suggest that autologous SC transplantation appears safe. No adverse effects were observed two years after SC injection [67,68,69,70,71,72,73]. Thus far, the practicability of autologous SCs for spinal cord injury has progressed through Phase 1 clinical trials [69,74]. For instance, adding SCs into the bioengineered PNS conduits increased regeneration by bridging nerve grafts. Santamaría et al. reported optimized safety and efficiency results of intralesional cell delivery for subacute human spinal cord injury, which allowed the development of new procedures for cell delivery into patients with chronic spinal cord injury. Key parameters of the transfer technique included accurate localization of the injury site, stereotaxic strategies to control needle trajectory, the process of entrance into the spinal cord, spinal cord motion reduction, the volume and density of the SC suspension, frequency of delivery, and regulation of shear stresses on cells [75]. Recent clinical trials have confirmed earlier reports on the feasibility and safety of autologous human SC transplantation in participants with chronic spinal cord injury [76].

It is crucial to notice that the efficacy of remyelination of injured CNS axons by grafted SCs has been reported to be limited because of different aspects of the hostile injury-induced microenvironment [77] and specific SC properties, like the weak migration of SCs into the CNS: SCs do not migrate extensively, display poor long-term survival, and remyelinate only some axons in the vicinity of the site of transplantation [78,79]. Astrocytes are known to limit the mobility of SCs in the CNS [80,81]. Another SC motility inhibitory mechanism may be the potential of myelin-associated glycoprotein (MAG) to suppress SC migration and induce SC death [82]. However, the regenerative ability of the nerve grafts could be enhanced by using genetically modified SCs expressing growth factors and other known neuroprotective and neuroregenerative molecules (Figure 2), like glial cell-derived neurotrophic factor (GDNF), neurotrophin-3 (NT-3), polysialyltransferase (PST), POU domain class 6 Homeobox 1 (POU6F1), etc. [83,84,85]. For instance, CNS migration of retrovirally transfected SCs expressing the polysialylated form of the neural cell adhesion molecule (NCAM) was significantly increased without altering their differentiation program or impairing their myelinating ability [86]. Furthermore, the grafting of these SCs significantly enhanced functional recovery after spinal cord injury [87]. Similar results were obtained with SCs retrovirally transduced to overexpress the cell adhesion molecule L1, which is known to support neurite outgrowth and is implicated in the myelination process. Transplantation of these SCs enhances early events in spinal cord repair after injury, as evidenced by faster locomotor recovery in mice [88].

Interestingly, experimental data suggest a higher efficacy of a combination strategy with SC transplantation, like, for instance, the paired elevation of cyclic AMP (cAMP) levels or paired administration of neuroprotective agents or growth factors, which improves the effect of SCs after spinal cord injury [89,90,91,92]. Similarly, amplified neurotrophin secretion by the implanted SCs, virally transfected to produce a bifunctional neurotrophin molecule D15A, which mimics both NT-3 and brain-derived neurotrophic factor (BDNF), alone or together with chondroitinase ABC resulted in a significantly augmented number of axons in the contusion site of the injured spinal cord, promoting axonal regeneration and locomotion [93,94].

Furthermore, the suspension of SCs in bioactive matrices can improve the longevity of transplanted SCs and upregulate their ability to assist axonal regeneration in the injured spinal cord [95,96]. In the model of complete spinal cord transections, the administration of SCs and Matrigel, or SCs and a fibrous piezoelectric polyvinylidene fluoride trifluoroethylene conduit, resulted in the detected regeneration of brainstem axons [95,97]. Co-transplantation of SCs and other cell types, such as bone marrow mesenchymal stromal cells or neural stem cells (NSCs), is also considered a more effective therapy than transplantation of SCs alone following spinal cord injury [38,40]. In addition, SC transplantation as a treatment for spinal cord injury can be supported by combination with electroacupuncture. Tan et al. reported that electroacupuncture improves the survival and proliferation of transplanted SCs, inhibits SC apoptosis, and increases the expression of neuregulin 1 (NRG1) type III in SCs [96]. These effects were associated with promoting axonal regeneration, altering astrogliosis by inducing the extension of astrocytic processes at the SC graft interface, and facilitating remyelination after spinal cord injury [96,97].

Although our understanding of the molecular mechanisms by which transplanted SCs employ a reparative effect on the injured spinal cord is quite limited, the proven clinical benefits include improvement of axon regeneration, remyelination of newborn or sparing axons, control of the inflammatory response, and preservation of the survival of damaged tissue [40,73]. Therefore, multiple barriers still must be overcome, including standardization of human SC manufacturing and understanding of their longevity in the injured tissue after transplantation, to enable feasible and successful treatments for spinal cord injury patients. Finally, despite the established clinical and safety improvements in human SC transplantation for spinal cord injury resulting from recent advances in understanding SC biology and improvements in SC cultures from human nerves, SC transplantation as a treatment strategy for peripheral polyneuropathies has not been systematically followed.

### 2.2. Schwann Cells for Treating Non-Traumatic CNS Diseases

Various human neurological disorders, such as multiple sclerosis, Parkinson’s disease, Huntington’s disease, or Alzheimer’s disease, result from damage or loss of neurons and glial cells in the brain or spinal cord. Demyelinating CNS diseases can arise due to traumatic insults, genetic mutations, metabolic defects, autoimmune reactions, hypoxic conditions, infectious diseases, or exposure to inflammatory mediators or toxins [98]. These conditions disrupt axon–glial interactions and often lead to secondary demyelination. However, in multiple sclerosis, for instance, demyelination is a primary event, and axonal myelin-preservation signals are still reported. Furthermore, hereditary myelinating defects originate from mutations in structural myelin genes or associated transcription factors.

Various demyelinating disorders are accompanied by continuous and disabling myelin loss [99]. CNS demyelination often results in severe functional impairment, which explains a great interest in designing treatments that promote repair in the CNS and patient recovery. Remyelination is a main issue for avoiding neurodegeneration and irreversible failures of function [100]. Like spinal cord injury models, the involvement of SCs in CNS remyelination during CNS inflammation has also been reported. In a model of inflammatory degenerative CNS neuropathy in dogs, dedifferentiated or injury-responsive SCs were detected in cerebral and cerebellar white and gray matter lesions, as well as in the brainstem, which was associated with a local reduction in axonal damage [101]. A similar response of SCs to axonal damage was described in the canine distemper virus model of multiple sclerosis [102]. Fascinatingly, examination of SC remyelination in the rodent models of ethidium bromide-induced areas of demyelination in the spinal cord following endogenous and transplant-mediated remyelination revealed the CNS, rather than PNS, origin for most SCs that remyelinate damaged CNS axons [103].

This evidence of gliogenesis in the lesioned CNS is of special clinical importance, as it highlights the role of regeneration-promoting cells that can be engaged for therapeutic purposes. Cell replacement therapy is a promising approach for myelin repair. While various experimental studies demonstrated the convincing repair potential of grafted myelin-forming cells, such as OPC, olfactory ensheathing cells (specialized glial cells of the CNS and PNS showing the characteristics of astrocytes and SCs), and neural and embryonic stem cells, SCs are among the most promising contenders for autologous transplantation [37]. Such conclusions are further supported by recent data indicating that glial cells may play a crucial role in non-cell autonomous neurodegeneration in neurological disorders, contributing to glial crosstalk and its impact on neurons. Particular emphasis has been given to glial cells in neurologic autoimmune disorders because autoantibodies, in addition to targeting neurons, may also target glial cells, causing a disparity in neural networks and synaptic homeostasis, as well as stimulation of neuroinflammation [104,105].

There are more than 30 identified autoimmune diseases of the nervous system, mediated, in general, by CNS- or PNS-targeting antibodies [106]. Autoimmune disorders of the CNS include demyelinating disorders, such as multiple sclerosis and neuromyelitis optica, as well as paraneoplastic and autoimmune encephalomyelitis. In addition to genetic predisposition, the breakdown of immune tolerance and the progression of autoimmune responses may be triggered by infections. For instance, the results obtained in some multiple sclerosis models suggest that prolonged Epstein–Barr virus (EBV) or cytomegalovirus (CMV) infections may induce autoreactive memory T cells that can be reactivated after CNS injury [107]. New data have added strong evidence that EBV infection may cause MS [108,109]. One possible mechanism is molecular mimicry, where EBV protein sequences resemble myelin and other CNS proteins, triggering an autoimmune response against myelin and CNS antigens. Other possibilities may include a dysregulated immune response toward EBV, leading to enhanced bystander activation and an inadvertent autoreactive response, or infection-induced brain tissue damage, resulting in an immune response to immune-privileged tissue antigens.

Cell replacement therapy with normal or genetically engineered SCs delivered to the diseased or injured brain may provide the foundation for developing efficient new treatment approaches for various human neurological diseases (Figure 2).

#### 2.2.1. Multiple Sclerosis and Schwann Cells

Multiple sclerosis is a chronic, demyelinating disease of the CNS characterized by inflammation, demyelination, and axonal damage resulting from the infiltration of immune cells into the CNS and the autoimmune targeting of oligodendrocytes and myelin. The pathological characteristic of multiple sclerosis is the existence of focal demyelinated lesions detected in the CNS [110]. Although spontaneous remyelination might happen in multiple sclerosis, as was demonstrated by the analysis of postmortem tissues from multiple sclerosis patients [111,112], its efficiency, or remyelination index, varies substantially among patients and correlates with functional outcome [113]. Even though accessible therapies can control multiple sclerosis inflammation, they have little effect on progressive neuropathology. In addition, none of the pharmacological agents tested for improving intrinsic myelination pathways delivered reliable termination of demyelination or myelin repair [114,115]. At the same time, the results of transplanting myelin-forming cells have presented convincing evidence for the concept that prompt remyelination protects axons from degeneration and fosters morphological and functional recovery in multiple sclerosis [100,115].

Earlier data demonstrating that spinal cord multiple sclerosis lesions often contain regenerating myelin sheaths formed by SCs [80,116,117] have been confirmed later [118], supporting the potential of SCs to contribute to CNS repair in multiple sclerosis. Interestingly, SCs have also been observed in the CNS of neuromyelitis optica patients [119]. Together, this highlights a critical need for feasible pro-remyelinating strategies to advance tissue repair and reduce long-term axonal damage. SC transplantation has been discussed as a promising approach that may aid clinical recovery in individuals with multiple sclerosis [120].

Preclinical studies using experimental autoimmune encephalomyelitis (EAE) rodents showed that transplantation of olfactory unsheathing cells and SCs into the demyelinating area markedly augmented the repair and regeneration of myelin [37,121,122]. For instance, allografted SCs not only survived and migrated within the parenchyma but also intermingled with astrocytes in demyelinated lesions, aligned with axons, and formed P0 protein internodes, i.e., remyelinated lesions in the inflamed CNS. Noticeably, this was associated with a marked decrease in animal mortality [123]. These experimental results align with early data indicating SC involvement in CNS remyelination in EAE animals [49,124]. Notably, accelerating remyelination in the EAE model is sufficient to maintain axonal integrity and neuronal function [125].

New data indicate that triggering receptor expressed on myeloid cells-2 (TREM2), detected on microglia, regulates the clearance of myelin debris and remyelination [126]. TREM2 agonistic antibodies promoted myelin uptake and degradation in a model of multiple sclerosis, supporting remyelination and axonal integrity [127]. Interestingly, SCs also express TREM2, which plays a critical role in the energy metabolism of SCs and in assisting nerve regeneration [128]. Thus, TREM2 agonists may be potential therapeutic tools for targeting SCs or improving SC-based therapies in different demyelinating neuropathies.

Much less is known about the therapeutic potential of SCs for other CNS autoimmune diseases, such as neuromyelitis optica spectrum disorders or myelin-oligodendrocyte glycoprotein antibody-related diseases, which manifest as optic neuromyelitis, acute demyelinating encephalomyelitis, or cortical encephalitis. This is despite the involvement of CNS glial cells in their pathogenesis having been proven, suggesting that they may be potential clinical targets for innovative treatment approaches [129,130]. Regardless, inhibiting demyelination and promoting myelin sheath regeneration are crucial therapeutic strategies for treating autoimmune CNS diseases.

Thus, while the existence of SCs in the CNS in specific pathophysiological settings has been repeatedly confirmed [115], the signals that hinder or permit SCs to take control of central axon remyelination are not yet wholly understood [131]. Identification of the CNS factors that regulate SC longevity and motility in the central parenchyma, such as the astrocyte-derived molecules or myelin components, for instance, is crucial to design reasonable treatment approaches for the utilization of SCs in CNS repair to complement oligodendrocyte remyelination and to reduce axonal loss.

#### 2.2.2. Parkinson’s Disease and Schwann Cells

Parkinson’s disease is a common neurodegenerative disorder characterized pathologically by intracytoplasmic inclusions or Lewy bodies composed of α-synuclein aggregates in neurons.

The oxidant–antioxidant theory, as one of several hypotheses proposed to explain the etiology of Parkinson’s disease, asserts that oxidative damage triggered by reactive oxygen species (ROS) can promote dopaminergic neuron degeneration [132]. Accumulating results revealed that oxidized lipids, proteins, and nucleic acids accumulate in the brain tissues of patients with Parkinson’s disease, suggesting that free radicals may affect both α-synuclein aggregation and disease progression [133]. Although a few pathogenic pathways have been implicated in Parkinson’s disease, including mitochondrial dysfunction and oxidative stress, altered iron metabolism and ferroptosis, failure of the ubiquitin–proteasome system, and impaired protein clearance, the processes of apoptosis and/or autophagy, and neuroinflammation, their interaction, and cellular involvement are not yet entirely understood [134]. Neuroinflammation is a notable pathological indication of Parkinson’s disease, and reactive microglia could scavenge abnormal α-synuclein produced by neurons.

The loss of peripheral nerve fibers and activation of SCs with the increased expression of inflammatory cytokines IL-1β, IL-6, and TNF-α were also observed in Parkinson’s disease patients, demonstrating the potential involvement of SCs in the pathophysiological process of the disease [135,136]. Remarkably, α-synuclein immunoreactivity was detected in SCs near spinal nerve roots [137], and phosphorylated α-synuclein deposits were seen in SCs [138]. New results confirmed that phosphorylated α-synuclein presence in vagus nerve SCs caused activation of the TLR2/MyD88/NF-κB signaling pathway, leading to neuroinflammatory responses and Parkinson’s disease autonomic dysfunction [139,140,141]. These data suggest that abnormal SC α-synuclein accumulation may be an additional pathogenic biomarker for Parkinson’s disease and a new target for autonomic dysfunction in Parkinson’s disease by blocking the SC phosphorylated α-synuclein/TLR2 signaling pathway. Furthermore, in animal studies of Parkinson’s disease, it has been reported that glucagon-like peptide-1 (GLP-1) receptor agonists, such as exendin-4, provide neuroprotective effects in the brain—defense of dopaminergic neurons in the substantia nigra, block of dopamine loss in the basal ganglia, and preservation of motor control [142]. Remarkably, exendin-4 has been reported to support the survival, proliferation, and migration of SCs, as well as myelination, in the DRG neuron/SC co-culture system [143,144]. One clinical trial in patients with Parkinson’s disease revealed that exendin-4 had beneficial effects on cognition and memory [145].

Importantly, the therapeutic capability of SCs for Parkinson’s disease has also been examined in experimental animal models, based on their proven neuroregenerative properties in the PNS [146]. For instance, in the 1-methyl-4-phenyl-1,2,3,6-tetrahydropyridine- and 6-hydroxydopamine (6-OHDA)-induced models of Parkinson’s disease, it was reported that PNS grafts could survive in the CNS, increase dopamine concentrations in the brain, and improve behavioral tests in rodents, which was associated with the secretion of basement membrane components by SCs [147,148]. Next, the transplantation of GDNF- or FGF-expressing SCs and nigra grafts in 6-OHDA-lesioned rats markedly improved the survival of dopaminergic neurons [149,150]. Moreover, SCs grafted into the brains of 6-OHDA-induced Parkinson’s disease in mice diminished the damage to dopaminergic neurons [151]. The results of SC co-drafting studies were also confirmed in a primate model of Parkinson’s disease [152]. Notably, one human clinical trial investigates whether autologous SCs can be used to restore damaged dopaminergic neurons in the CNS [146]. Thus, intracerebral cell-based therapy by administration of autologous PNS-derived SCs is an encouraging therapeutic approach that may prevent or inhibit Parkinson’s disease progression. However, further clinical studies are needed to confirm these initial findings.

#### 2.2.3. Alzheimer’s Disease and Schwann Cells

Alzheimer’s disease is a neurodegenerative disorder manifested by extracellular amyloid beta plaques in the grey matter, intraneuronal hyperphosphorylated tau filaments, neuronal death, synapse elimination, and brain atrophy, and is associated with free radical and oxidative stress, metabolic dysregulation, and upregulated pro-inflammatory cytokines [153]. For instance, upregulated generation of reactive oxygen and reactive nitrogen species and defective mitochondrial dynamic balance can lead to the misfolding of amyloid-beta and hyperphosphorylation of tau proteins, affecting tau protein kinase activation and phosphatase inhibition, thus causing the formation of neurofibrillary tangles, a hallmark of Alzheimer’s disease [154]. Dysfunctional mitochondria may further release ROS, increase the production of pro-apoptotic proteins, and cause apoptosis of neurons [155].

Recent clinical data emphasize myelin destruction as a key and particularly aggravating factor in disease pathogenesis [156]. Oligodendrocyte death and demyelination are thought to happen secondary to neurodegeneration in Alzheimer’s disease [157]. Although the results of many studies support the concept that the transplantation of stem cells, which may transdifferentiate into neuronal lineage, could improve synaptic plasticity, cognitive performance, and neurogenesis circuitry in Alzheimer’s disease models [158,159], SC-based treatment approaches have not been widely evaluated.

Yan et al. investigated the effect of transplanting SCs into the brains of rats with Alzheimer’s disease in a model [160]. First, they demonstrated that, in the in vitro co-culture system, SCs significantly promoted NSC proliferation, differentiation, and synaptic formation compared to nerve growth factor (NGF) alone. In vivo experiments revealed that the transplantation of either NSCs alone or (NSCs + SCs) reduced glial fibrillary acidic protein (GFAP) and S100β but increased choline acetyltransferase mRNA expression and markedly endorsed learning and memory in affected rats. However, the effect of (NSCs + SCs) brain co-grafting was significantly more robust. The authors concluded that co-transplantation of NSCs and SCs extensively lowered astrocyte counts and enhanced cholinergic neuron counts, while accelerating the recovery of learning and memory function in Alzheimer’s disease rats compared to grafting NSCs alone [160]. Notably, these results were partially confirmed when it was reported that co-culturing SCs and NSCs supported the differentiation of NSCs into neurons and the secretion of higher concentrations of neurotrophic factors, such as BDNF and glial cell line-derived neurotrophic factor (GDNF) [161].

Furthermore, in preclinical animal studies of Alzheimer’s disease, it has been reported that GLP-1 receptor agonists, like exendin-4, liraglutide, and lixisenatide, cross the blood–brain barrier and provide neuroprotective effects in the CNS—reduction of β-amyloid plaques, oxidative stress, and the chronic inflammatory response, and averting damage of synapses and memory [142]. Interestingly, exendin-4, a GLP-1R agonist, is known to support the survival, proliferation, and migration of SCs, as well as myelination, in the DRG neuron/SC co-culture system [143,144]. The recent development of micromolecular synthetic neurotrophin mimetics, selectively acting on neurotrophin receptors, offers a unique opportunity for novel therapeutic approaches to neurodegenerative diseases [156]. For instance, BDNF mimetics, known to mediate pro-myelinating effects in the nervous system, may have beneficial therapeutic potential for Alzheimer’s disease [156]. Finally, TREM2 is known to be involved in the amyloid pathology of Alzheimer’s disease through the regulation of glial cell proliferation, homing, and phagocytosis of amyloid plaques [162]. TREM2 agonists have been shown to enhance microglial cell survival and function, offering a potential avenue for developing TREM2-targeting therapies for Alzheimer’s disease [163]. At the same time, TREM2 is highly expressed in SCs, regulating SC metabolism and myelinating function [128]. This may provide an additional indication of further evaluation of modified SC-based treatments in Alzheimer’s disease.

However, many more experimental studies are needed to prove the feasibility of using SCs for mono- or combinational treatment of Alzheimer’s disease.

#### 2.2.4. Huntington’s Disease

Huntington’s disease, the common autosomal-dominant neurodegenerative disease, presents with severe motor, cognitive, and psychiatric symptoms. A mutation of the *HTT* gene, which is associated with the synthesis of a toxic, misfolded huntingtin protein, leads to specific pathological features such as neuronal dysfunction and subsequent cell death [164]. Stem cell therapy, as a new treatment for Huntington’s disease, utilizes different types and sources of stem cells, such as mesenchymal stem cells (MSCs), NSCs, embryonic stem cells (ESCs), and reprogrammed stem cells or induced pluripotent stem cells (iPSCs) [165]. Stem cells can potentially replace lost neurons, improve neuron regeneration, and deliver pro-survival factors. Another advantage of autologous stem cell transplantation is its origin from the patient, which alleviates the risk of immunological rejection and averts the need for immunosuppression therapy [166]. Interestingly, new animal studies have demonstrated that Huntington’s disease mutation present in iPSCs can be corrected before the transplantation of these cells, without limiting their successful differentiation into neurons [167].

Several ongoing clinical trials investigate the safety and efficacy of autologous stromal cells in patients with Huntington’s disease [168]. Although no data have revealed the effect of SC transplantation in patients with Huntington’s disease so far, many new encouraging results can predict the approach of a new era of Huntington’s disease therapeutics.

#### 2.2.5. Leukodystrophies

Leukodystrophies are heterogeneous neurodegenerative diseases that primarily affect myelination in the CNS. However, some forms of leukodystrophy are characterized by progressive demyelination of both the CNS and PNS, causing severe neurological symptoms [169]. For instance, axonal degeneration in both the CNS and PNS has been described in Krabbe disease (globoid cell leukodystrophy) and metachromatic leukodystrophy. At present, more than 30 forms of leukodystrophies have been categorized as caused by mutations in myelin- or oligodendrocyte-specific genes or in other white matter components, including astrocytes, neuronal axons, microglia, and brain vasculature [170].

While the results from animal models and human preclinical studies suggest that glial cell therapy may serve as a cell-based remedy for glial-driven leukodystrophies in the near future, no data have demonstrated the feasibility of using SCs for these diseases [171]. However, new studies may accelerate the development of this approach.

In metachromatic leukodystrophy, a mutation in the *ARSA* gene, encoding the lysosomal enzyme arylsulfatase A, results in the accumulation of sulfatides in multiple cell types, including SCs, leading to cytotoxicity and demyelination of most of the nerve fibers of the CNS and PNS [172]. Metachromatic material, accumulating in peripheral nerves in patients with metachromatic leukodystrophy, consists of SCs and endoneural macrophages loaded with typical lysosomal inclusions of sulfatides or inclusion bodies. SCs and macrophages die, and demyelination of myelin in the PNS follows [173]. These are also morphological alterations of the endoplasmic reticulum and mitochondria in SCs. Several types of inclusion bodies were seen in SCs on sural nerve biopsy. In addition, the endoplasmic reticulum, mitochondria, and lysosomes in SCs demonstrated significant ultrastructural alterations, suggesting that a subcellular metabolic insufficiency of SCs in metachromatic leukodystrophy may be responsible for the PNS pathology [174]. Similarly, “abnormally dense mitochondria with thickened cristae” in SCs were reported in another ultrastructural analysis of nerve biopsies from a patient with metachromatic leukodystrophy [175].

Elevation of sulfatide synthesis in oligodendrocytes and SCs of ARSA^−/−^ mice has been reported to significantly upregulate the accumulation of sulfatides and the development of myelin pathology in the CNS and PNS [176]. The sulfatides are structurally normal but disrupt myelin metabolism.

New data on modeling mitochondrial metabolism in metachromatic leukodystrophy showed that human SCs transfected to overexpress ROS in response to sulfatides and displaying bioenergetic and mitochondrial defects could be treated with metformin, which repaired their metabolic activity and diminished ROS production [177]. Interestingly, analysis of myelination in mouse axons by SCs transplanted from human sural nerves of healthy donors and metachromatic leukodystrophy patients revealed the formation of metachromatic granules within SCs in the leukodystrophic grafts [178]. This demonstrated that the grafted SCs remained ARSA deficient during nerve regeneration and were unable to utilize the enzyme produced by the surrounding cells. At the same time, fibroblasts isolated from a metachromatic leukodystrophy patient and genetically modified to overexpress ARSA were able to competently transfer ARSA to oligodendrocytes and SCs through the inner membrane in vitro [179], suggesting a few therapeutic directions.

In the second classic genetic leukodystrophy—globoid cell leukodystrophy, or Krabbe disease, there are genetic defects in a lysosomal hydrolase, galactosylceramidase, which catabolizes the myelin sphingolipid galactosylceramide. This causes reactive astrocytic gliosis, the rapid and nearly complete disappearance of myelin and myelin-forming oligodendrocytes and SCs, and infiltration of macrophages (“globoid cells”) [180]. A progressive dysfunction of SCs was detected in the animal model of Krabbe disease [181,182].

While bone marrow or hematopoietic stem cell transplantation and mesenchymal stem cell therapy have proven the feasibility and potential effectiveness in leukodystrophies, SC-based approaches have not yet been developed [171,183,184]. However, in one study, the clinical effect of mesenchymal stem cell infusion in metachromatic leukodystrophy patients has been explained by the mesenchymal stem cell differentiation into SCs in vivo [185].

#### 2.2.6. Schwann Cells in Amyotrophic Lateral Sclerosis

Amyotrophic lateral sclerosis (ALS), or Lou Gehrig’s disease in the USA, is a terminal neurodegenerative disorder and the most common form of motor neuron disease. Although approximately 50 potentially pathogenic and several causative genes have been identified in patients with ALS, the etiology of ALS appears heterogeneous and is not well understood [186,187]. The neuropathological signature of ALS is characterized by the loss of neuromuscular connections, axonal retraction, and the progressive degeneration of both upper motor neurons and lower motor neurons in the CNS, accompanied by astrogliosis and microgliosis [188]. Emerging evidence has demonstrated common extra-motor involvement in ALS, implicating the PNS as a converging point. An increasing amount of data reveal the presence of sensory and autonomic dysfunction in ALS, which has been reported in up to 30% of cases [189]. For instance, 80% of ALS patients displayed a drop in intraepidermal nerve fiber density on skin biopsy [190]. Similar signs of axonal degeneration and peripheral motor and sensory dysfunction have been reported in ALS animal models [191,192]. Thus, it is not surprising to read “ALS is a well-known peripheral neurodegenerative disease” [193].

However, the pathogenic significance of SCs in distal motor axonopathy is poorly understood. In ALS patients, motor axon loss in the peripheral nerves causes a Wallerian-like degeneration characterized by denervated SCs and immune cell infiltration. Clinical and preclinical results suggest that peripheral axons are lost before the death of cell bodies in the CNS [194,195]. However, peripheral nerve pathology in ALS demonstrates a chronic and progressive degenerative and inflammatory phenotype. It is possible that chronically activated SCs also trigger or orchestrate an inflammatory mechanism in ALS.

Immunohistochemical analysis of SCs on sections of the sciatic nerves from ALS subjects demonstrated a substantial enhancement of GFAP and S100b staining compared with sections from control nerves, suggesting a significant SC reactivity [196]. Differential GFAP and S100β staining in SC subsets resembled denervated or repair phenotypes, such as those observed in Wallerian degeneration. New data confirmed that cultured SCs prepared from sural nerve biopsy material obtained from an ALS patient appeared senescent; however, when treated with exosomes isolated from a cadaver donor SC culture, their growth potential in vitro was significantly improved [197]. A recently identified subset of SCs expressing *Adamtsl1*, *Cldn14*, and *Pmp2* that preferentially myelinate large-caliber motor axons has been reported to be reduced in both ALS model mice and ALS patient nerves [198,199].

Sciatic nerves from superoxide dismutase (SOD1)^G93A^ rats, resembling ALS, also showed that S100b^+^ and GFAP^+^ SCs correspond to two different—myelinating and denervated—cell populations [196]. Cytoplasmic and mitochondrial SOD1 convert superoxide radicals into molecular oxygen and hydrogen peroxide, and mutations in SOD1, associated with alterations in SOD1 functionality and/or aberrant SOD1 aggregation, contribute to ALS pathogenesis. Rodent transgenic ALS models, developed by expressing a human *SOD1* transgene with ALS-associated mutation G93A, reproduce the major phenotypic features of human ALS [200]. Importantly, the development of motor neuron disease is not due to a loss of SOD1 function, as many mutant forms of SOD1 retain nearly normal or even elevated SOD1 activity [201,202]. Thus, while neither upregulated expression of wild-type SOD1 nor obliteration of endogenous SOD1 caused motor neuron disease, it is evident that the disease is induced by an acquired toxicity of mutant SOD1 independent of its dismutase activity [203].

Increased SOD1 activity can ameliorate the production of ROS in the mitochondrial intermembrane space, contributing to mitochondrial damage in SOD1^G93A^ rats [204]. Therefore, mechanisms critical for ALS progression may include mitochondrial dysfunctions, excitotoxicity, oxidative stress, and changed Ca^2+^ metabolism. The effects of SOD1 G93A mutation in non-neuronal cells, such as glial cells, in ALS may be associated with the altered redox balance and perturbed expression of Ca^2+^ transporters that may be responsible for altered mitochondrial Ca^2+^ fluxes [205].

Interestingly, sciatic nerves from ALS rats also showed upregulated endoneurial expression of colony-stimulating factor 1 (CSF1) and IL-34: CSF1 was expressed by a subset of phagocytic S100^+^ p75^NTR+^ SCs engulfing myelin debris, while IL-34 was expressed by a subset of denervated GFAP^+^ SCs with the morphology of repair SCs, which also expressed c-Kit (stem cell factor, SCF, receptor). In the sciatic nerves of ALS donors, CSF1 and IL-34 were detected in elongated SC-like cells [196]. Most importantly, CSF1-R and c-Kit receptor inhibitor masitinib prevented SC reactivity in degenerating nerves and ameliorated sciatic nerve pathology in ALS rats.

Furthermore, ALS-derived mouse SCs did not affect 3D axonal migration in vitro, whereas it was strongly enhanced by normal SCs [206]. Analysis of ALS mice demonstrated that disease onset is associated with a significant initiation of GFAP expression in SCs, suggesting their distress [207]. iNOS immunoreactivity was also increased in SCs on peripheral nerves in SOD1 mutant mice [208]. SCs expressing a dismutase active mutant SOD1 were shown to reduce disease progression in ALS mice, as the removal of mutant SOD1 reduced survival. This suggests a link between slow disease progression in ALS mice and a protective impact of dismutase active mutant SOD1 in SCs [209]. Importantly, SCs transfected with mutant SOD1 expressed low levels of peroxiredoxin 1, and the expression of peroxiredoxin 1 mRNA was significantly decreased in the lumbar spinal cord of SOD1^G93A^ mice [210]. Peroxiredoxin 1, an antioxidant molecule derived from SCs, protected motor neurons from hydrogen peroxide-induced cell death, suggesting that the reduction of peroxiredoxin secreted from SCs contributes to increased ROS and accompanies motor neuronal death in ALS [210]. Similarly, peroxiredoxin 6 secreted by SCs has been shown to significantly inhibit neuron apoptosis and improve neurological recovery in different models [211].

However, while myelinating SCs may attempt to remyelinate wounded axons, the ongoing motor neuron degeneration in ALS overcomes this regenerative competence, leading to axonal nerve dysfunction [212]. In addition, the loss of myelinating SCs, which are prone to apoptosis, as observed in the SOD1-G93A/TDP-43 model [213], also decreases the overall support available to motor neuron axons [187]. Dysfunctional SCs in the SOD1-G93A model display modified gene expression, low axonal metabolic support, and the failure to remove debris from degenerating axons [214].

Non-myelinating SCs may also participate in muscle weakness and atrophy associated with the peripheral nerve dysfunction seen in ALS patients. Evidence shows that Remak SCs are demoted in the SOD1-G93A model of ALS, suggesting that Remak SC dysfunction may precede or accompany motor neuron degeneration [198]. It has also been reported that the denervation status correlates with the S100^+^ terminal SC loss from the neuromuscular junction in the Thy1-hTDP-43WT mouse model of ALS [213]. Terminal SCs are non-myelinating SC types that are crucial for synapse development and maintenance [215]. Early and permanent functional alterations of perisynaptic SCs in the neuromuscular junction were also detected in the SOD1-G37R mouse model of ALS, recapitulating many aspects of the human disease [216,217]. Significantly, similarly to axonal SCs, perisynaptic SCs contribute to neuromuscular junction reinnervation following denervation by adopting a repair phenotype [218], which may be regulated by muscarinic acetylcholine receptors (mAChRs) [212]. Comparative transcriptomic analysis of perisynaptic and other SCs concluded that synaptic SCs may be responsible for age-associated degeneration of the neuromuscular junction [219].

An analysis of terminal SCs in mice expressing the G93A-SOD1 mutation allowed the conclusion that these SCs may play a role in motor terminal degeneration and denervation in motor neuron pathology [220]. Disease progression is associated with a loss of complete S100 and P75NTR immunoreactivity in denervated neuromuscular junctions, suggesting a loss of perisynaptic SCs. This was confirmed in different animal models and human biopsies [220,221,222]. Interestingly, it has been suggested that perisynaptic SCs may play a pathogenic role in ALS [187]. This might be mediated by a compromised synthesis of the neuromuscular junction regulator agrin, or secretion of matrix metalloproteinase (MMP)-degrading agrin [223], or generation of CXCL12α (stromal cell-derived factor, SDF-1) in perisynaptic SCs [224], or SC-induced Connective Tissue Growth Factor-related pathways [225].

Together, these results support the notion that SCs are promising targets for therapeutic intervention in ALS. Thus, despite attracting limited consideration, SC abnormalities in ALS hold potential for innovative therapies targeting pro-myelinating pathways. As mentioned above, c-Kit signaling may drive some SC abnormalities in ALS [196]. Masitinib, an inhibitor of the protein tyrosine kinase c-kit and other receptors, lowered SC reactivity and immune cell infiltration in nerves in SOD1-G93A rats [196]. Phase 2b/3 and Phase 3 clinical trials involving ALS patients treated with masitinib demonstrated a slowing of disease progression rates [187,226]. In line with this is a first case report of an 81-year-old patient with rapidly progressive ALS who was treated with weekly intravenous infusions of allogeneic SC-derived exosomes to potentially restore impaired SC and motor neuron function [197]. The results revealed a trend for clinical stabilization during the infusion period, suggesting an interesting approach to address damaged SCs and probably neuropathy in patients with ALS.

## 3. Schwann Cells in PNS Pathology

Peripheral nerve disorders can be genetic or acquired as a result of traumatic, toxic, metabolic, infectious, or autoinflammatory conditions. As non-myelinating and myelinating SCs ensheathe the axons of peripheral nerves, playing a critical role in PNS development, physiology, and regeneration, the loss or dysfunction of SCs is an etiologic or pathogenic feature of many diseases and conditions that compromise the PNS (Figure 1 and Figure 2). Understanding the interplay between SC plasticity/polarization and peripheral neuropathies could reveal novel strategies for treating and managing sensory, motor, or mixed PNS pathology.

### 3.1. Schwann Cells for Treating PNS Injury

PNS injuries, because of traction, contusion, compression, ischemia, total/partial section, or complications of infection, disorders, or toxicity, often promote sensory and motor disability and neuropathic pain, representing many cases of chronic disability in younger and otherwise healthy individuals [227]. Biological therapy, utilizing various cellular, molecular, and bioengineering approaches, may be the solution for enhancing the outcome of peripheral nerve repair after surgery or rehabilitation [228,229].

Given their role in the pathophysiological repair response and remyelination after peripheral nerve injury, the application or targeting of SCs for improved outcomes has been considered for many years. For example, Guenard et al. in 1992 transplanted SCs into a sciatic nerve defect in rats and revealed improved axonal regeneration and myelin formation [230]. Later, the clinical application of SCs from different sources and SC-like cells for accelerating nerve regeneration was repeatedly tested and characterized [19,229,231,232]. The first human experience with autologous SCs in 2016–2017, to supplement nerve repair after complete transection of sciatic nerves by a boat propeller injury and after partial damage of the sciatic nerve by a gun wound of the leg, revealed that patients regained complete or partial sensory and motor function recovery [233,234]. New studies now focus on the use of allogeneic stem cell-derived SCs and SC-like cells to promote axonal regeneration and repair peripheral nerve injury-induced neuropathy [235].

These data demonstrate that cell replacement-based therapeutic strategies utilizing exogenous SCs can be curative [236]. There are two common cell replacement approaches for peripheral nerve repair: the isolation of repair SCs for cell transplants and the controlled differentiation of pluripotent stem cells or the lineage conversion of available somatic cells into induced SCs [237]. However, experimental and clinical data demonstrate that direct SC transplantation frequently results in relatively low therapeutic efficacy, particularly for long-distance peripheral nerve lesions. This is because cell administration may induce cell damage and instant loss in the circulatory system. The interaction between SCs and the extracellular matrix also plays a crucial role in peripheral nerve repair [238]. In addition, while SCs are susceptible to local milieu signals due to their plasticity, they cannot maintain their repair phenotype for a prolonged period because the signals gradually alter [239]. Given the limited repair ability of SCs for extended peripheral nerve defects, it is essential to complement SCs through gene engineering, co-transplantation with other cells, or by combining them with bioactive or structural scaffolds [240]. Thus, specific biomaterials have been engineered as functional carriers to mimic the compositional and topological signs of the extracellular matrix (ECM) and maintain the biological properties of SCs for effective cell transplantation [239].

Ultimately, a comprehensive understanding of myelin clearance pathways is crucial for identifying cellular mechanisms that can be targeted to promote myelin clearance and enhance remyelination and recovery after PNS injury. Using the SC conditional knockout of the calcineurin B model, Reed et al. showed that calcineurin may regulate autophagy in SCs and play an important role in myelin clearance after injury [241]. Likewise, NGF could initiate autophagy in dedifferentiated SCs, upregulate clearance of myelin debris, and promote axon and myelin regeneration at an early stage of peripheral nerve injury [242].

These data provide novel insights into the mechanisms of pharmacological therapy for activating SC autophagy after peripheral nerve injury, offering a potential avenue for its preclinical testing and verification.

#### 3.1.1. Combinational Schwann Cells Transplantation for Treating PNS Injury

Different biomaterials that mimic the natural regeneration environment may protect SCs and help cell integration into the injured CNS or PNS during SC transplantation. Such biomaterials should be biocompatible, biodegradable, and permeable, allowing cell migration, adhesion, and encapsulation. Various natural biomaterials like fibrin, chitosan, collagen, laminin, and fibronectin, as well as synthetic biomaterials like poly (glycolic acid), polylactic acid, and poly (lactic acid–glycolic acid) copolymerization (PLGA), offer feasible cell-to-cell interaction and outstanding physical properties [243,244]. SCs can be resuspended in a pre-gel solution for in situ gelation or pre-encapsulated in a hydrogel to prevent serious cell damage during administration. For example, collagen hydrogel can provide crucial extracellular matrix molecules and simulate the nanofibrous structure of native nerves to support SC functions. It has been shown that a tethered type-1 collagen gel helps maintain the expression of growth and guiding factors from SCs and promotes nerve regeneration in a rat sciatic nerve defect model [245]. Chitosan conduits combined with collagen, laminin-, or fibronectin-based matrices significantly improved the therapeutic effects of SCs in bridging long nerve gaps in different animal models [246,247].

The role of plasma fibrin in peripheral nerve regeneration has also been demonstrated that the blood–nerve barrier can be disrupted after nerve injury, inducing fibrinogen leakage into the wounded area and cleavage by thrombin to form fibrin cables. These fibrin cables may stimulate SC polarization to a regenerative phenotype and provide physical guidance for SC migration [248,249]. Schuh et al. used a fibrin hydrogel as a transplantation platform for SCs and reported that they mixed fibrin with collagen to create an engineered neural tissue. They proved that adding fibrin aided marked improvement of SC survival and endorsed nerve growth both in vitro and in vivo when compared to collagen alone [250]. Polysaccharides, like hyaluronic acid- and alginate-based hydrogel that can host exogenous SCs, have been used as bridging materials for peripheral nerve injury repair [239]. An alternative to natural biomaterials is synthetic hydrogels, which utilize polyethylene glycol, poly(2-hydroxyethyl methacrylate), and poly[N-(2-hydroxypropyl)-methacrylamide]. However, they are not frequently used alone as SC carriers [239].

Curcumin, the main active ingredient in turmeric from the root of *Curcuma longa*, may accelerate the repair of sciatic nerve injury in rats by lowering SCs apoptosis and increasing their proliferation and myelination [251,252]. Interestingly, the addition of curcumin-encapsulated chitosan nanoparticles to the neural guidance conduit prepared by poly-L-lactic acid (PLLA) and surface-modified multi-wall carbon nanotubes (MWCNTs) and filled with SCs significantly improved the regeneration and functional recovery of injured peripheral nerves [253]. Resveratrol, a natural plant compound, can promote sciatic nerve crush injury recovery by promoting the autophagy of SCs [254]. Interestingly, to address local hypoxia in the early stages of peripheral nerve injury, Ma et al. used perfluorotributylamine-based oxygen-carrying fibrin hydrogel in combination with SCs and concluded that enhanced survival of SCs accelerated axonal regeneration, remyelination, and recovery of the motor and sensory function of the regenerating nerves [255].

Recent experimental animal and preclinical human studies revealed that the transplantation of SCs in combination with axon guidance channel technology, like nerve scaffolds, e.g., decellularized nerve conduits, promotes the repair of injured peripheral nerves and thus represents an applicable strategy for the management of peripheral nerve injury [256,257]. Many studies have reported the development of different types of conduits to increase the therapeutic efficacy of SC transplantation to repair peripheral nerve injuries [240,258]. For instance, the rate and distance of axonal regeneration across a 10 mm nerve defect were increased when SCs were added to a synthetic matrix peptide hydrogel [259]. Increased efficacy of peripheral nerve regeneration, using an electrically conductive biodegradable porous neural guidance conduit, produced from PLLA, MWCNT, and gelatin nanofibrils coated with erythropoietin-loaded chitosan nanoparticles, for transplantation of allogeneic SCs, has also been reported [260]. Similarly, using collagen tubes confirmed that adding SCs to a guidance channel markedly increased the gap space that can be repaired after extended segmental defects of peripheral nerves [247]. Huang et al. have developed a composite nerve conduit with a PLGA hollow tube as the outer layer and gelatin methacryloyl encapsulated with vascular endothelial growth factor (VEGF)-A transfected SCs as the inner layer [261]. This approach provided a comprehensive solution for improved angiogenesis and nerve regeneration.

A rat model of laryngeal nerve injury was used to determine the efficacy of SCs and neural stem cells co-transplantation with a laminin–chitosan–PLGA nerve conduit to repair 5 mm long recurrent nerve injuries [262]. The results demonstrated that the combination approach promoted nerve regeneration markedly better than the autograft method, the use of SC/nerve conduit only, or the neural stem cells/nerve conduit only approaches [263]. Remarkably, SC-aligned scaffolds demonstrated better regeneration success in a 15 mm nerve defect in the rat model than MSC-aligned constructs [246]. A novel PLGA copolymer foam conduit of longitudinally aligned 60–550 µm channels and luminal surfaces promoting SC adherence, when mixed with SCs and implanted across a 7 mm gap in the rat sciatic nerve, demonstrated a strong regenerative capability and may be used to promote precisely guided neural regeneration [264]. Similar results were reported after testing SC-encapsulated chitosan–collagen hydrogel nerve conduits [265]. Treatment of PLGA nerve guidance conduits seeded with SCs and implanted into a 10 mm sciatic nerve gap with low-intensity ultrasound resulted in larger areas of axonal myelination and improved histological recovery [266]. Remarkably, low-intensity ultrasound is known to upregulate the proliferation of SCs and the expression of myelinating genes in SCs [267].

Altogether, experimental and clinical data suggest that SC engrafting and targeting for peripheral nerve injury is a promising regenerative strategy that may restore neurological function (Figure 1). However, special attention should be given to evaluating SC myelination and axonal growth outcomes of scaffold-based therapies for peripheral nerve injury in experimental animal models [268]. It is also essential to notice that SCs play a fundamental role in the selectivity of nerve regeneration: mature SCs exhibit modality-specific phenotypes and mechanisms, allowing them to differentially control the selective regeneration of motor and sensory axons [269].

#### 3.1.2. Modified Schwann Cells for Treating PNS Injury

Transplantation of modified or engineered SCs has emerged as a promising cell-based therapy to facilitate peripheral nerve recovery and remodel the microenvironment, thereby maintaining nerve homeostasis. One of these promising approaches is the genetic engineering of SCs, which will be reviewed later. Several examples of other therapeutic SC modifications are presented here.

Magnetic actuation induced by applying superparamagnetic iron oxide nanoparticles (SPIONs) to SCs can also encourage and preserve repair-like phenotypes of SCs, which is associated with the promotion of nerve regeneration and functional recovery in rat sciatic nerve injury models [270]. In the same model, short-term low-frequency electrical stimulation upregulated BDNF expression in the injured sciatic nerves, inducing earlier SC myelination. It elevated the number of myelinated fibers and the myelin sheath thickness [271]. Brief low-frequency electrical stimulation may enhance regeneration even after delayed nerve repair [272]. It is known that SCs respond to electrical stimulation by increasing the proliferation and expression of NGF, GDNF, and BDNF, which can be associated with the augmented outgrowth of injured axons in vivo [273]. A magnetic force-based mechanical stimulation of SCs demonstrated a compelling ability to enhance neurite outgrowth in vitro and nerve regeneration in vivo [274].

Furthermore, new data revealed that the three-dimensional aggregation of SCs upregulated their acquisition of a repair phenotype and that SC spheroid-derived secretome promoted neurite outgrowth in dorsal root ganglion (DRG) neurons [275]. Transplantation of preassembled SC spheroids in rats with a sciatic nerve transection improved the restoration of injured nerves and motor functional recovery.

Notably, recent reports have demonstrated that inhibiting pyroptosis in SCs, induced by PNS injury, promotes the healing of the sciatic nerve and the restoration of motor function in rats [276]. Pyroptosis is a peculiar form of programmed cell death that could hinder peripheral nerve regeneration. It was reported that silencing Rab32, a small GTPase protein family member, using an adeno-associated virus diminished nerve injury-induced SC pyroptosis and stimulated peripheral nerve regeneration.

Another agent affecting SCs and supporting PNS regeneration is melatonin. It could promote the expression of Parkin, an E3 ubiquitin ligase that mediates mitophagy—an important cytoprotective mechanism of the removal of impaired mitochondria. Melatonin has been shown to suppress SC apoptosis, reduce the production of mitochondrial ROS, and promote myelin regeneration and peripheral nerve repair [277]. Suppression of ROS production by Parkin-mediated mitophagy can reduce SC death associated with peripheral nerve injury [277]. Similarly, it was reported earlier that melatonin improved the proliferation and migration of SCs during peripheral nerve injury, thus advancing nerve regeneration [278,279]. SCs, expressing melatonin membrane receptors MT1 and MT2, respond to melatonin via a complex signaling cascade, stimulation by upregulated expression of GDNF, dedifferentiation, migration, and reduced glial scar formation [280]. Selective activation of cannabinoid receptor 2 (CB2) may also be considered as a therapeutic approach to promoting the remyelination process following peripheral nerve injury. Pharmacological activation of CB2 by specific agonists upregulates the expression of pro-myelination-related genes, including Sox10, Egr2, and Tprv4, in SCs, and alters the expression of Myelin protein zero (MPZ) and myelin basic protein (MBP). In the sciatic nerve injury model, activation of CB2 increases SC proliferation and pro-myelination, improves myelin thickness, and accelerates the remyelination of injured peripheral nerves. At the same time, genetic ablation of SC-specific CB2 intensified pain and motor dysfunction in this model [281]. The authors concluded that SC-expressed CB2 is essential for peripheral nerve regeneration.

Shen et al. have recently constructed zero-dimensional Black phosphorus quantum dots modified with the antioxidant β-carotene and evaluated their potential for peripheral nerve repair [282]. These quantum dots activated the PI3K/Akt and Ras/ERK1/2 signaling pathways in SCs, thereby affecting their pro-inflammatory potential and promoting the functional recovery of neurons by enhancing axon remyelination and regeneration, as well as facilitating intraneural angiogenesis in rat and dog models of peripheral nerve injury.

These and other experimental data hold great potential for enhancing the effectiveness of cell delivery and the subsequent therapeutic outcomes of SC-based transplantation approaches in stimulating peripheral nerve regeneration.

### 3.2. Schwann Cells as Targets and Tools for Non-Traumatic Peripheral Neuropathy

Peripheral neuropathy of different etiologies, including trauma and side effects of diseases and treatments, is a frequently encountered neurological problem that leads to sensory and motor disorders (Figure 1). The treatment of these conditions primarily focuses on specific clinical symptoms and surgical interventions. Despite the latest advancements in special medications and surgical techniques, functional improvement remains disappointing. Thus, developing, designing, and evaluating novel treatment approaches for peripheral neuropathies is well justified.

#### Schwann Cells in Diabetic Neuropathy

Diabetic neuropathy, one of the most common complications of diabetic patients, is a unique neurodegenerative disease of the PNS that differently targets sensory axons, autonomic axons, and later motor axons. The number of diabetic neuropathy cases has tripled globally since 1990, suggesting that it is the fastest growing of all neurologic conditions [283]. Although it is not initially thought of as demyelinating neuropathy, SCs are affected by chronic hyperglycemia, and more severe cases exhibit features of demyelination [284,285]. Advanced glycated end products, inflammation, and oxidative stress can damage SCs, decrease the production of NGF, and thus provoke axonal degeneration [286,287,288,289]. For instance, SCs, damaged by oxidative stress-induced mitochondrial dysfunction, demonstrated significantly decreased viability and increased apoptotic death associated with Bcl2, NF-κB, mTOR, and Wnt signaling [290,291]. A specific form of metabolic remodeling, involving aberrant ketogenesis within SCs, has been recently reported in streptozotocin-induced type I diabetes mellitus [292]. This maladaptive peripheral ketogenesis depends on cannabinoid type-1 receptor signaling, as its silencing or pharmacological inhibition rebalances SC metabolism, reduces histopathological abnormalities, and improves neuropathic symptoms.

In sural nerve biopsy samples from patients with diabetes and progressive worsening of neuropathy, the full range of ultrastructural abnormalities, as well as reactive, degenerative, and proliferative changes in SCs, have been repeatedly reported [293]. Segmental demyelination, abnormal myelin sheaths, and remyelination have been demonstrated in various animal models of diabetic neuropathy [294,295,296,297]. High glucose significantly upregulated the expression of peroxide and superoxide, as well as the pro-inflammatory cytokines IL-1β, IL-6, and tumor necrosis factor alpha (TNF-α), in cultured SCs through the TLR4/NF-κB pathway [298]. Significantly, targeted activation of the liver X receptor signaling pathway or specific inhibition of the ROS-producing enzyme NADPH oxidase-4 in vivo and in vitro reduces diabetes-induced ROS production in SCs, thereby reversing the functional alterations of peripheral nerves and restoring the homeostatic profiles of MPZ and peripheral myelin protein 22 (PMP22) [299]. Targeting the peroxisomal biogenesis pathway in SCs with, for instance, palmitic acid may also be an effective strategy for preventing and treating diabetic neuropathy [300].

Although the pathogenesis of diabetic peripheral neuropathy is multifactorial, numerous data suggest an important role of SCs in the pathogenesis of diabetic neuropathy [288,289]. Evaluating the transcriptional profiles and intercellular communication of SCs in the nerve microenvironment of a mouse prediabetes and neuropathy model, Eid et al. demonstrated that neuropathic SCs develop a pro-inflammatory and insulin-resistant phenotype under prediabetic conditions [301]. The results are not limited by SC dysfunction, such as impaired paranodal barrier function, injured myelin, decreased antioxidative capacity, and declined neurotrophic support for axons [302]. For instance, thioredoxin-interacting protein, which is associated with oxidative stress and inflammation, may mediate SC dysfunction in diabetic peripheral neuropathy via PI3K/Akt pathway-mediated autophagy and apoptosis [303]. Collecting evidence proves that SC death due to high glucose is a part of the pathological process of diabetic peripheral neuropathy [287,304,305]. Bax and cleaved caspase-3 levels were significantly higher, while Bcl-2 levels were reduced considerably in SCs cultured in a high-glucose medium [298]. Interestingly, the overexpression of nuclear factor erythroid 2-related factor 2 (NRF2) in SCs inhibited hyperglycemia-induced SC apoptosis by modulating the TLR4/NF-κB signaling pathway [298]. High glucose-induced ferroptosis of SCs has also been recently demonstrated by inhibiting the NRF2 signaling pathway [306]. As oxidative and endoplasmic reticulum stress; nitrification; autophagy; and NRF2, PI3K/AKT, ERK, and Wnt/β-catenin signaling are involved in the pathological pathways of high glucose-induced SC apoptosis, verification of the therapeutic targeting of these mechanisms in SCs should help identify practical approaches to treating diabetic peripheral neuropathy.

Alkaloids and polysaccharides from Aconite (Fuzi), a well-known traditional Chinese medicinal herb used for the treatment of diabetes and paralysis, were effective in accelerating the nerve conduction velocity in diabetic rats and protected SCs against high-glucose injury by decreasing superoxide anion and peroxide levels and regulating Bax, Bcl-2, cytochrome C, caspase-3, and caspase-9 pathways [307,308]. Jatrorrhizine, a protoberberine plant alkaloid with anti-inflammatory activity, may enhance SC myelination in diabetic mice by controlling the NRG1–ErbB2–PI3K–AKT pathway [309].

Next, as shown above, NRF2 expression may reduce cell death in SCs under hyperglycemic conditions [298]. In the streptozotocin-induced diabetic mouse model, trehalose, a naturally occurring disaccharide, prevented nerve conduction velocity deficits and sciatic nerve damage [310]. It also blocked high glucose-induced oxidative damage and apoptosis in SCs, thereby reducing the high glucose-induced expression of pro-apoptotic mitochondrial proteins, including Bax, Puma, Bak, and Bim. In a similar model in rats, exendin-4, a synthetic glucagon-like peptide-1 receptor agonist, inhibited peripheral nerve degeneration in association with GLP-1 receptor activation, anti-apoptotic effects, and restoration of cAMP levels in SCs [143].

Some data showed that a 6-week oral treatment with minocycline, a known inhibitor of SC activation, significantly improved peripheral and autonomic neuropathy in type 2 diabetic patients [311]. Interestingly, a new study found that bupropion, an aminoketone antidepressant, reduced glucotoxicity in SCs and blocked SC apoptosis and myelin damage in treated hyperglycemic mice [312].

Based on data showing that neurotrophins, erythropoietin, neurosteroids, and ascorbic acid may prevent and reduce behavioral, electrophysiological, and pathological changes in polyneuropathy animal models, it has been suggested that SCs could be considered therapeutic targets for hormones and vitamins [313]. Erythropoietin decreased intracellular ROS levels in SCs in high-glucose culture conditions, increased cell viability, and reduced the apoptotic rate of SCs [314,315]. NGF has been reported to block sciatic nerve degeneration and demyelination in diabetic rats and significantly reduce high glucose-induced endoplasmic reticulum stress and subsequent apoptosis in SCs via the PI3K/Akt/GSK3β and ERK1/2 signaling pathways [316]. Bone morphogenetic protein 5 (BMP5) may also enhance diabetic peripheral neuropathy by enhancing mitochondrial function and reducing apoptosis in SCs [317]. Interestingly, taurine, a natural semi-essential amino acid, can improve the morphology of the damaged myelin sheath and restrain SC apoptosis in the sciatic nerve of diabetic rats, which presumably was mediated by upregulated expression of NGF and phosphorylation levels of Akt and GSK3β in SCs [318]. Additionally, histone deacetylase inhibitors can increase BDNF expression in SCs, thereby ameliorating diabetic peripheral neuropathy [319]

Furthermore, the transplantation of neural crest-like cells, differentiated into SC-like cells, into diabetic mice effectively improved the impaired vascular and neuronal functions [320]. Similarly, stem cells from various sources, such as bone marrow-derived cells, endothelial progenitor cells, and mononuclear cells, are capable of successfully reversing the symptoms of diabetic polyneuropathy due to their neuroprotective effects and neovascularization [321]. For example, the administration of SC-like cells, prepared from human tonsil-derived mesenchymal stem cells, in BKS-db/db mice (a type 2 diabetes model) may alleviate diabetic neuropathy through the remyelination and recovery of sensory neurons [322]. Notably, the impact of different cell therapies on SC function in diabetic neuropathy has been repeatedly reported, including decreased SC apoptosis, promotion of SC proliferation and viability, and increased myelin formation and myelin-related protein expression, as well as restoration of myelination [235]. An improved understanding of the role of SC plasticity in diabetes should lead to new strategies for treating and managing diabetic peripheral neuropathy [323,324]. For instance, immortalized SC lines have been suggested as valuable tools for therapeutic approaches to diabetic peripheral neuropathy [325].

### 3.3. Schwann Cells in Immune-Mediated Neuropathies

Inflammatory or immune-mediated neuropathies refer to heterogeneous disorders with pathogenic immune cell infiltration of peripheral nerves, accompanied by demyelination and axonal degeneration (Figure 1) [326]. By networking with immune cells, SCs contribute to the creation of immune responses that may lead to inflammatory neuropathies, such as autoimmune demyelinating diseases. The clinical picture ranges from Guillain–Barré syndrome (GBS) to chronic inflammatory demyelinating polyneuropathy (CIDP). GBS and CIDP are two of the most common inflammatory diseases of the PNS [326]. Acute inflammatory demyelinating polyneuropathy (AIDP), as the GBS subtype, is highly prevalent in the USA and Europe. Although the origin of the autoimmunity in many autoimmune demyelinating diseases has not been precisely elucidated, clinical and experimental data provide clear evidence that the immune response, facilitated partly by SCs, often has a destructive role and promotes disease development both in inflammatory neuropathies and in many hereditary neuropathies [327,328]. SC expression of molecules associated with the antigen processing and presenting machinery, as well as TNF-α and IL-1, is upregulated in inflammatory conditions [13]. On the other hand, SCs can become targets of an autoimmune response in certain inflammatory neuropathies, like the subtypes of GBS (Figure 1).

Recent data suggest that during the segmental demyelination with live myelin-forming SCs, macrophage infiltration is responsible for the myelin removal by its phagocytosis. Importantly, macrophage-attacked SCs transdifferentiated into inflammatory demyelinating SCs with a distinctive demyelination pathology—myelin uncompaction [34,329]. This pathway may represent a novel autophagy-mediated myelin clearance mechanism by SCs in inflammatory demyelinating neuropathies [330]. These findings provide an important basis for a better understanding of the pathobiology of demyelinating peripheral neuropathies and the development of novel diagnostic and therapeutic modalities.

It is important to mention that peripheral inflammatory neuropathy might develop in a small cohort of cancer patients treated with immune checkpoint inhibitors targeting CTLA-4 or PD-1/PD-L1 signaling in T cells. The reported neuropathy cases commonly resemble acute or chronic inflammatory demyelinating or vasculitic neuropathies [331,332,333,334]. Acute neurotoxicity is also documented in up to 30% of cancer patients during CAR-T-cell therapy, which is associated with “immune effector cell-associated neurotoxicity syndrome” or cytokine release syndrome [335,336]. A case of chronic inflammatory demyelinating polyneuropathy developed in a patient with metastatic malignant melanoma after treatment with interferon-α has also been reported [337]. Additionally, paraneoplastic neuropathies are also immune-mediated due to the immune reaction to malignant cells expressing onconeural antigens and mimicking molecules expressed by neurons.

#### 3.3.1. Schwann Cells in Guillain–Barré Syndrome

GBS and its AIDP variant are well-known post-infectious immune-mediated polyneuropathies and the representatives of autoimmune neuroinflammatory diseases of the PNS. In many cases, axonal and demyelinating GBS variants arise after infection and implicate molecular mimicry mechanisms relying upon the structural similarity between pathogens and heterogeneous host autoantigens [338,339]. An important pathophysiological pathway, thus, is complement-mediated nerve damage caused by autoantibodies targeting the peripheral nerves, with anti-ganglioside and anti-galactocerebroside antibodies being the most common in GBS [129,340]. Approximately 60% of patients with GBS demonstrate anti-ganglioside antibodies in their sera [341]. Notably, anti-GM1, -GM1b, -GD1a, and -GalNAc-GD1a antibodies are often detected in GBS patients after *Campylobacter jejuni* infection; anti-GM2 antibodies after CMV infection; and anti-galactocerebroside antibody after *Mycoplasma pnuemoniae* [342]. In Miller Fisher syndrome, a GBS variant, 80–90% of patients display antibodies to GQ1b that target cranial motor nerves and may be associated with molecular mimicry after *Campylobacter jejuni* infection [343,344]. The elevated levels of pro-inflammatory cytokine IFN-γ and TNF-α expression correlated with extreme clinical severity, whereas the anti-inflammatory IL-4, IL-10, and TGF-β correlated with recovery [345]. New results demonstrated that blood and CSF autoreactive CD4^+^ and CD8^+^ T cells recognize peripheral nerve myelin antigens P0, P2, and PMP22 in patients with GBS [346].

Up to 25% of GBS patients have circulating autoantibodies recognizing SCs and binding to the distal tips (leading lamella) of SCs [347]. However, another study reported that 13% of GBS patients showed strong IgG reactivity against SCs [348]. In addition, the deposition of activated complement components has been reported on the surface of SCs in AIDN patients [349]. These and other reports advocate that SCs can serve as targets of immune-mediated demyelination [350]. In experimental autoimmune neuritis, an animal model of GBS, an activated complement complex can be detected on SCs before myelin degradation, suggesting the role of this pathway in demyelination [351]. In GBS, autoantibodies to MPZ, P2, gliomedin, and ganglioside GM1, expressed by SCs, may elicit the complement cascade and SC lysis. Notably, the demyelinating variants of GBS are the consequence of SC and myelin sheath injuries. Analysis of axonopathy pathways in transgenic mice expressing GM1 either solely in neurons or glia confirmed that the roles of both primary and secondary axonal injury are mediated by complement-fixing anti-GM1 ganglioside antibodies that target peripheral nerve axonal and SC membranes, respectively [352].

On the other hand, complement components and complement regulatory molecules can be produced and expressed by SCs [12,350,353,354], suggesting an important immune regulatory function of SCs in immune-mediated neuropathies. The immunomodulatory activity of activated SCs also plays a substantial role in GBS pathogenesis, as well as other infection-associated PNS inflammation. This may be the result of either indirect pathogen activation of cellular constituents in peripheral nerves, e.g., SCs, or direct SC infection [355]. Crucial to such a response are SCs, as they are the providers of pivotal components of the adaptive and innate immune reactions against invading pathogens. For instance, infection of human SCs with alphaviruses, known to be associated with neurological complications, caused a significant alteration of the expression of genes encoding different immunoregulatory factors in SCs, signifying their potential role in the pathogenesis of GBS after alphavirus infection [356]. Furthermore, SCs in sural nerve biopsies from GBS patients demonstrated upregulated expression of molecules associated with the antigen processing and presenting machinery [13]. Finally, animal models of GBS have demonstrated that perisynaptic SCs, rather than macrophages and neutrophils, are the main cellular population responsible for clearing axonal debris following distal nerve damage [357].

Although the role of SCs in the neuroinflammatory pathogenesis of GBS is well established, the full scope of potential targeting of SCs or the feasibility of their therapeutic application in GBS remains to be fully deciphered. New data may accelerate these developments.

#### 3.3.2. Schwann Cells in Chronic Inflammatory Neuropathies

The most common forms of chronic inflammatory neuropathies are chronic inflammatory demyelinating polyradiculoneuropathy, multifocal motor neuropathy, and polyneuropathy associated with monoclonal gammopathy of unknown significance (MGUS). CIDP, as a demyelinating autoimmune disorder in the PNS, is characterized by symmetric, progressive limb weakness and sensory loss. CIDP is closely related to AIDP and may be considered its chronic counterpart. Related autoantibodies commonly damage the myelin sheath in the PNS, causing segmental demyelination and remyelination: more than 40% of patients with CIDP display antibodies targeting components of myelinated nerves [358]. Circulating IgG autoantibodies recognizing proliferating, non-myelinating SCs were detected in more than 25% of CIDP patients [347]. Antibodies against integral peripheral myelin components, such as MPZ, PMP2, and PMP22, are also relatively common [359]. Another report confirmed that a high percentage of CIDP patients presented a strong IgG or IgM reactivity against the myelin components, although the patterns of IgG or IgM fixation on myelinating SCs were different, suggesting that discrete myelin antigens may be recognized by autoantibodies [360]. Myelin in peripheral nerves expresses LM1. Gangliosides GM1 and GD1b are expressed in higher quantities in motor and sensory nerves, respectively. GM1 is also widely expressed on SC microvilli [361]. Autoantibodies binding to LM1, GM1, GD1b, and ganglioside complexes containing LM1 have been recognized in patients with CIDP [344]. Circulating IgM anti-GM2 antibodies were detected in patients with multifocal motor neuropathy with an earlier disease onset [362]. They also specifically target SCs and activate complement, similarly to IgM anti-GM1 on motor neurons.

New evidence suggests that immune recognition of additional antigens expressed in the noncompact myelin may also be involved in the pathogenesis of CIDP [363]. Specifically, identifying IgG4 antibodies against the node of Ranvier structure in patients with CIDP represented a significant advance in the field [358]. Importantly, these findings suggest that IgG4 antibodies with paranodal or nodal reactivity are pathogenic, with minimal contribution from inflammatory reactions, immune cells, or complement.

SCs may play a role in antigen presentation in inflammatory neuropathies. Analysis of the expression of co-stimulatory molecules on SCs in different types of immune-mediated neuropathies revealed the constitutive expression of the co-stimulatory molecule BB-1 on unmyelinating SCs in all studied nerves. However, prominent upregulation of BB-1 expression on the myelinating SCs was detected only in CIDP [364]. Of note, SCs expressing BB-1 also expressed HLA-DR. Furthermore, in CIDP patients’ biopsies, but not those from healthy controls, SCs expressed the adhesion/T-cell stimulatory molecule CD58 (lymphocyte function-associated antigen 3, LFA-3) [365].

Finally, SCs in CIPD may lose their plasticity and neuroregenerative potential. Human and animal SCs treated with serum from CIDP patients display imbalanced expression of BDNF, GDNF, and NGF [366]. The fact that SC abnormalities were preventable by granulocyte-macrophage colony-stimulating factor (GM-CSF), which is typically lower in CIDP sera, suggests its role in facilitating the loss of SC growth support in CIDP and may open up novel therapeutic interventions.

#### 3.3.3. Monoclonal Gammopathy-Associated Peripheral Neuropathies

Plasma cell dyscrasia and other hematological disorders may be associated with a variety of neuropathies of both axonal and demyelinating types. This includes anti-MAG neuropathy, POEMS (polyneuropathy, organomegaly, endocrinopathy, M protein, skin changes) syndrome, and immunoglobulin light-chain amyloidosis, all of which exhibit characteristics of paraneoplastic disorders. Anti-MAG and light-chain amyloidosis can cause painful sensory neuropathy, often accompanied by autonomic features, whereas POEMS syndrome can result in axonal, demyelinating, or mixed neuropathy [367,368].

Neuropathies were also observed in lymphoma, MGUS, Waldenstrom macroglobulinemia, and multiple myeloma. Pathogenesis is likely a direct effect of monoclonal immunoglobulins on the peripheral nerve, leading to a demyelinating process. Multiple myeloma may induce various paraneoplastic neuropathies, including sensory–motor, axonal, and demyelinating neuropathies. In paraproteinemia, monoclonal IgG or IgA often triggers axonal lesions, while IgM induces demyelinating problems. Monoclonal IgM deposits are noticed in the widened lamellae of myelin fibers, and myelin debris is included in SCs [369]. Waldenstrom macroglobulinemia can result in peripheral neuropathy in up to 47% of patients, who mostly complain of sensory loss [370]. Predominant motor neuropathies are less common and associated with a high titer of IgM antibodies that recognize ganglioside GM1 and MAG [371]. Amyloidosis is associated with up to 40% of neuropathies in myeloma, specifically, with free λ light chains [334].

Neurological complications—sensory, axonal, peripheral neuropathies or sensory and motor multiple mononeuropathies—have been described in 20 percent of cryoglobulinemia cases [372]. Cryoglobulinemic neuropathy is also referred to as paraneoplastic, with typical monoclonal IgM precipitation [334]. However, the role of SCs in the pathogenesis of hematological disease-associated neuropathies has not yet been established. A comprehensive understanding of SC involvement in immune-mediated nerve damage is critical and would be constructive in developing potential targeted therapies.

### 3.4. Schwann Cells in Infection-Induced Demyelination and Neuropathy of the PNS

Neurotropic viruses that can infect the CNS and cause encephalitis, inflammatory immune disorders, or meningitis are well characterized and include herpes simplex virus (HSV)-1/2, CMV, varicella-zoster virus (VZV), Ebola virus, and rabies virus, and several Arbo-, entero-, and Picornaviruses [373,374]. EBV, HSV-1, and herpesvirus have been linked to demyelinating diseases of the CNS [375,376]. Demyelination is among the most noticeable neurological consequences of SARS-CoV-2 infection in both the CNS and PNS [377]. As discussed above, GBS and CIDP are major PNS demyelinating diseases induced by viral infection or abnormal immune system function. The pathogenesis of the infection-induced demyelination may range from direct infection and lysis of glial cells with degeneration of myelin to immune damage of myelin or supporting cells by cell- or antibody-mediated immune cytotoxicity due to viral antigen presentation on infected cells, myelin epitope spreading into the circulation, or molecular mimicry between virus and myelin antigens.

#### 3.4.1. Schwann Cells in Infection-Associated Neuropathies of the PNS

Various viral infections have been associated with peripheral demyelinating disease, including infections with cytomegalovirus, EBV, human immunodeficiency virus (HIV), and VZV [378,379,380,381]. HSV and VZV have been demonstrated to infect SCs in vitro; however, the clinical significance of this pathway for direct SC infection remains to be proven [355]. SCs are known to be involved in the pathogenesis of certain infectious peripheral inflammatory neuropathies [355]. However, whether the immune reaction is mediated by intracellular SC infection or SC activation via pattern recognition receptors or pro-inflammatory cytokines and chemokines remains unclear. For instance, infection of human SCs with alphaviruses, known to be associated with neurologic complications, caused a significant alteration of expression of genes encoding different immunoregulatory factors in SCs, signifying their potential role in the pathogenesis of GBS after alphavirus infection [356].

A common source of distal peripheral neuropathy is HIV [382]. The involvement of SCs in HIV-associated distal sensory polyneuropathy has also been reported. The HIV-1 envelope glycoprotein gp120 can activate SCs to release pro-inflammatory cytokines and chemokines, thereby mediating neurotoxicity [383]. Deterioration of small myelinated and unmyelinated nerve fibers during HIV infection might be linked to activated SCs and attracted infected macrophages [384].

Persistent infection of immature SCs with lymphocytic choriomeningitis mammarenavirus blocked the ability of differentiated cells to form compact myelin sheaths without causing SC apoptosis or cytopathic defects [385].

Chronic infection with *Trypanosoma cruzi* (Chaga’s disease), an intracellular protozoan parasite able to invade different cell types, including SCs [386], is associated with degeneration of parasympathetic ganglia and partial functional loss of autonomic nervous system innervation. *T. cruzi* expresses a trans-sialidase/parasite-derived neurotrophic factor to mimic mammalian neurotrophic factors and bind to tyrosine kinase receptors to invade SCs [355,386].

*Corynebacterium diphtheriae*, also known as the Klebs–Löffler bacillus, is a pathogenic bacterium that causes diphtheria. The diphtheria toxin alters protein function in the host by inactivating the elongation factor EF-2. The diphtheria toxin is known to damage SCs both in vitro and in vivo [387,388]. Diphtheric neuropathy is characterized by vacuolation and fragmentation of myelin sheaths in the PNS. Secreted protein exotoxin could gain access to endoneurial fluid, bind to the SC membrane receptor, and catalyze ADP-ribosylation and inactivation of an elongation factor required for SC protein synthesis. Elongation factor 2 (EF-2), a member of the GTP-binding translation elongation factor family, permits the transfer of the peptidyl tRNA/mRNA complex from the ribosome to the peptidyl site during protein synthesis. Inhibition of myelin synthesis can lead to diphtheritic polyneuropathy [389,390].

Thus, since peripheral neuropathy caused by pathogens is a significant source of clinical disability, new studies are required to determine whether therapeutic inhibition of SC infection and SC pro-inflammatory activation is effective in treating chronic infection-associated neuropathy [355].

#### 3.4.2. Leprosy and Schwann Cells

Leprosy, or Hansen’s disease, is a rare infection caused by obligate intracellular pathogens *Mycobacterium leprae* or *Mycobacterium lepromatosis* [391]. *M. leprae* has been shown to have a unique affinity for infecting and persisting within both non-myelinating and myelinating SCs, leading to PNS nerve injury, including demyelination and a loss of nerve function [392,393]. Neuropathy, including sensory loss or paresthesia, and related disabilities are the main medical concerns of leprosy. Many aspects of the pathogenesis of lepromatous neuropathy remain unclear, despite significant advancements in knowledge about the bacteriological, histopathological, immunological, and clinical features of *M. leprae* infection [394].

Intracellular *M. leprae* infection of SCs results in the subversion of glucose/lactate and lipid/cholesterol metabolism, mitochondrial dysfunction, expression of neurotrophic factors, dysbalanced production and secretion of cytokines, and upregulation of the expression of MMP and ECM proteins [393,395,396,397,398,399,400]. Interestingly, elevated expression of both pro-inflammatory TNF-α, IL-1, IL-6, IL-8 and anti-inflammatory TGF-β, IL-10 cytokines has been reported in SCs challenged with bacilli [393,401,402,403]. *M. leprae* also causes significant alterations in gene expression, leading to myelin dismantling, elevated SC plasticity, dedifferentiation, and proliferation [403,404,405].

Cytokines, chemokines, and MMP released from infected SCs and activated immune effector cells, as well as SC apoptosis induced by cytotoxic T cells recognizing *M. leprae* antigens expressed on SCs, trigger epithelioid granuloma formation and edema, ultimately prompting axonal atrophy and demyelination [400,406,407,408]. In addition to the standard therapeutic regimens for treating leprosy infection, new therapeutics for leprosy neuropathy are needed, and candidate vaccines have shown clear advantages in preventing nerve damage [394,409].

### 3.5. Schwann Cells in Inherited Peripheral Neuropathies

Inherited peripheral neuropathies comprise over 100 genetically defined disorders, and the majority of affected patients suffer from the demyelinating form of the disease, with a primary pathophysiology in SCs [410]. Other inherited peripheral neuropathy subgroups incorporate hereditary motor neuropathies and hereditary sensory and autonomic neuropathies, where motor and sensory neurons are affected. When SCs degenerate, such as in Charcot–Marie–Tooth disease type 1 (CMT1), peripheral nerves are demyelinated, and axons are injured (Figure 1). Clinically, this is characterized by progressive distal muscle weakness and atrophy, sensory deficits, loss of deep tendon reflexes, and reduced nerve conduction velocities.

#### 3.5.1. Schwann Cells in Charcot–Marie–Tooth Disease

Charcot–Marie–Tooth disease is a clinically diverse and genetically heterogeneous group of inherited peripheral nerve disorders with different primary cells involved in its pathogenesis and more than 1000 mutations in over 100 distinct CMT disease-associated genes [411]. Some CMT forms are frequently caused by mutations in SC-expressed genes [412,413]. CMT1 is triggered by myelinating SC dysfunction, and CMT2 results from axonal deficits. The most common form of CMT disease, CMT1A, is caused by a duplication of the gene encoding the peripheral myelin protein 22 (*PMP22*). Pathological overexpression of PMP22 mRNA and protein in SCs may be the principal mechanism of atypical myelin formation and axonal damage, as well as the abnormal trophic support required for the maintenance of axons [414,415].

CMT1B is an outcome of point mutations in the *P0* gene, encoding MPZ [114]. The periaxin gene encodes L- and S-periaxin, which are responsible for maintaining the stability of myelin in myelinating SCs and are mutated in CMT disease type 4F and Dejerine–Sottas syndrome, also known as CMT3 [416]. Duplication of specific segments containing the *PMP22* gene, a nearby gene *RAI1*, and sometimes additional genes, underlies many significant features of Yuan–Harel–Lupski syndrome, which is characterized by multiple neurological problems.

Common *PMP22* mutations affect PMP22 protein trafficking in SCs [417], which may be seen as cytoplasmic protein accumulation in SCs of the sciatic nerve in CMT patients and modeling mice [418,419]. These PMP22 ‘aggresomes’ may be toxic, as they can cause endoplasmic reticulum (ER) stress, SC apoptosis, and demyelination [420]. Interestingly, co-culturing PMP22 transgenic SCs with neurons induced abnormal SC differentiation, trafficking, and motility and impaired myelin formation [421]. New data have identified the first SC-specific protein, TMPRSS5 (spinesin), which is elevated in the blood of CMT1A patients and may provide a promising disease marker [422]. Transmembrane Serine Protease 5 demonstrated preferential expression in SCs, probably because of its regulation by the Sox10 transcription factor [423].

Utilizing CMT1A mice overexpressing human PMP22, Prior et al. detected that PMP22 decreased the expression of genes associated with lipid and cholesterol metabolism dose-dependently [424]. Similar lipidomic profiles and reduced expression of genes involved in lipid metabolism were confirmed in human CMT1A patients’ SCs prepared from iPSCs. This suggests that PMP22 regulates the lipid composition of the plasma membrane and lipid storage homeostasis in SCs in CMT disease. Thus, targeting the *PMP22* gene expression or lipid metabolism may hold some promise as a prospective therapeutic approach for CMT1A patients.

Progesterone, a neuroactive steroid, plays a crucial role in SC physiology through the classic intracellular progesterone receptors, membrane progesterone receptors, and gamma-aminobutyric acid (GABA) type A receptors. Steroids have been shown to affect SC morphology, proliferation, differentiation, motility, and myelination [425,426]. Progesterone may regulate myelin gene expression [427]. A progesterone antagonist has been shown to lower the overexpression of PMP22 and to improve clinical signs in *PMP22* transgenic rats [428]. Oral *L*-serine supplementation was tested in mice and humans. It decreased neurotoxic 1-deoxy-sphingolipid levels, which were associated with improved sensory symptoms and strengthening of the upper and lower extremities in patients with hereditary sensory and autonomic neuropathy [429,430]. Similarly, curcumin could improve clinical and neuropathological signs in the CMT mouse model by relieving ER stress and promoting SC differentiation [431,432].

In a proof-of-concept study, Passage et al. demonstrated a beneficial effect of high-dose ascorbic acid in an animal model of CMT1A [433]. Ascorbic acid-treated mice that overexpress the *PMP22* gene showed reduced pathology in sciatic nerves, improved behavioral measures, and an increased life span compared to untreated mice. Based on these encouraging findings, clinical trials testing ascorbic acid in CMT1A have been carried out, which, however, did not demonstrate promising results [313,434,435].

Neurotrophin 3, given in either a peptide form or via adeno-associated virus (AAV)-based gene transfer, demonstrated therapeutic efficacy in CMT [414,436,437]. NT-3 is a valuable autocrine agent supporting SC survival and differentiation in the absence of axons, which plays an essential role in the early stages of myelination associated with regeneration in the adult peripheral nerves [438].

The usefulness of stem cell therapy for CMT has also been demonstrated by treating C22 or Tr-J mice, a model for CMT1A, with human tonsil-derived mesenchymal stem cells differentiated into SC-like cells, which the authors refer to as neuronal regeneration-promoting cells [439,440]. The significant improvement in sciatic nerve regeneration and motor function and the increased number of myelinated axons following the transplantation of SC-like cells suggest the feasibility of this approach for clinical research on CMT1A.

Viral vectors have been employed in preclinical models for treating different CMT forms by expression of promising trophic factors, like NT-3, or by targeting the specific genes in neurons or SCs [441,442]. SC-targeting gene therapy is discussed below.

#### 3.5.2. Schwann Cells and Hereditary Neuropathy with Liability to Pressure Palsies (HNPP)

Deletion of one copy of the *PMP22* gene is the most frequent genetic cause of HNPP, a recurrent, episodic demyelinating neuropathy characterized by tingling, numbness, or loss of muscle function [443]. As a result of this deletion, the amount of PMP22 protein produced may be decreased by up to 50%. HNPP can also be caused by the *PMP22* gene mutations, leading to an unusually small and unstable protein production. The deficiency of the PMP22 protein affects the structure of myelin, impairing nerve transmission and leading to disruptions in nerve signaling [444]. New findings suggest that PMP22 dosage affects not only myelinating SCs but also non-myelinating SCs. Sural nerve biopsy of HNPP patients demonstrated increased unmyelinated axons in a single axon-containing non-myelinating SC subunit [445]. Focal thickening of the myelin sheath, seen as sausage-like swellings (tomacula) on nerve biopsy specimens, is a characteristic feature of HNPP [443].

Because numerous studies have revealed that the dosage of PMP22 may determine the type and severity of the related neuropathy, such as CMT or HNPP, treatment strategies should target the direct pathways affected by the *PMP22* gene dosage. As discussed above about CMT treatment, progesterone and ascorbic acid are known regulators of PMP22 expression in SCs. Gene replacement therapies are under evaluation for HNPP. High-throughput screening of cell lines expressing PMP22 reporters should facilitate the selection of novel candidates for pharmacological modulation of PMP22 expression. Another therapeutic strategy focuses on preserving SC-axonal interactions and their intimate connection by providing trophic factor support to degenerating axons in neurodegenerative diseases, including CMT and HNPP. This approach also involves manipulating SC-axonal signal transduction pathways (Figure 2 and Figure 3).

### 3.6. Schwann Cells and Other Types of Peripheral Neuropathy

Peripheral neuropathies may occasionally be associated with gastrointestinal diseases [446]. For instance, more than two-thirds of patients with inflammatory bowel disease endured axonal neuropathy with sensory predominance [447]. In ulcerative colitis, the most common neuropathology is acute inflammatory demyelinating polyradiculoneuropathy, whereas patients with Crohn’s disease often present with axonal motor and sensory neuropathy [448,449]. Neuropathy was detected in up to 23% of people with celiac disease, and these patients are at a three-fold increased risk of chronic inflammatory demyelinating neuropathy [446,450]. On the other hand, SC functions in digestive system diseases are well described [451]. Glial cells are crucial in maintaining the intestine’s physiological functions, including nutrient absorption, barrier integrity, and immune modulation [36]. It was also suggested that in the enteric microenvironment, the injury- and inflammation-associated upregulation of neurotrophic factors and cytokines may cause the transdifferentiation of mature enteric glia and recruitment and reprogramming of SCs, which then would regulate immune responses and perform neurogenesis [451,452]. Thus, targeting neurogenic mechanisms offers a promising approach to developing innovative strategies for acquired damage to the enteric nervous system. Notably, as an independent nervous system within the gastrointestinal tract, the enteric nervous system is a highly sophisticated neural network comprising over 100 million neurons and more than 400 million enteric glial cells [453].

Inflammatory neuropathies of the enteric nervous system are characterized by a dense infiltrate of immune cells associated with the neural microenvironment, leading to neuronal dysfunction and degeneration, sometimes including a complete loss of enteric neurons [454]. A recent study has reported a good example of therapeutic transplantation of SCs in gastrointestinal neuropathy. Pan et al. utilized an animal model of Hirschsprung disease—a congenital disorder characterized by the failure of the enteric nervous system to develop in the distal part of the intestine due to the failure of neural crest-derived precursors to colonize the developing intestine [455]. SCs were isolated, expanded in cultures, and transplanted to syngeneic animals with colonic aganglionosis. Results demonstrating the ability of SCs to engraft and restore contractile function in the aganglionic recipient smooth muscle suggest that extrinsic nerve-derived neuronal precursors could serve as an autologous source of neurons having the capability to regain innervation in the aganglionic bowel [456]. GDNF treatment of mice in a similar model stimulated gliogenesis and neurogenesis, which, at least in part, was due to SCs that functioned as stem cells [455].

These studies suggest that SCs associated with extrinsic gut innervation are a potential target for therapies aimed at restoring the enteric nervous system in the gut.

## 4. Peripheral Neuropathic Pain and Schwann Cells

Peripheral neuropathic pain is a pathophysiological condition caused by a primary lesion or dysfunction in the PNS. Neuropathic pain may be a consequence of trauma, viral infections, treatment, endocrine dysfunctions, cancer, or neurologic disorders, among others. Interestingly, pain is a serious and common problem in patients with multiple sclerosis, although the primary mechanism is still uncertain [457,458]. While multiple sclerosis is generally considered a CNS disease, some data indicate that PNS may also be involved [459,460].

Growing evidence suggests that non-neuronal cells, particularly glial cells, are also involved in the development and resolution of pain or pain syndrome. Specifically, SCs, by providing the immediate response to nerve trauma or injury, act as a key player in the induction and maintenance of neuropathic pain, and various receptors, ligands, ion channels, and factors expressed by SCs are involved in the regulation of different pain conditions [9]. For instance, the neurological manifestations of Fabry disease include both PNS and CNS involvement, with glycolipid deposits in SCs, DRG, and CNS neurons. Patients with Fabry disease experience chronic debilitating pain and peripheral sensory neuropathy [461]. New data have demonstrated that changes in signaling between SCs and sensory neurons may trigger peripheral sensory nerve dysfunction [462]. The authors found that Fabry SCs released an increased amount of S100 calcium-binding protein A10, also known as p11, which caused sensory neuron hyperexcitability.

PNS injury- or pathology-activated SCs have been shown to contribute to the increased neuronal activity that may result in chronic pain [463]. Under these conditions, SC-derived factors may enhance (inducible nitric oxide synthase (iNOS), nitric oxide (NO), iNOS/NO, TNF-α) or reduce (low-density lipoprotein receptor-related protein 1 (LRP1, CD91), erythropoietin, basal lamina components) pain sensitivity. Importantly, these pain-controlling effects of SCs can occur independently of demyelination or axonal degeneration [464]. For instance, activated SCs destroy the myelin sheath and produce factors that provoke hypersensitivity to pain, leading to a reduced pain threshold that triggers trigeminal neuralgia—a severe type of paroxysmal neuralgia with tender symptoms and sensations that are among the most painful [465]. Interestingly, the results of recent studies suggest that targeting TRPA1 channels in SCs may provide an innovative therapeutic approach to fibromyalgia-linked headaches [466,467].

Several pathways may be involved in SC-mediated neuropathic pain [465]. Following nerve injury, the activation of TNF-α receptors on SCs upregulates the expression of the P2X4 receptor, an ATP-gated cation channel that mediates Ca^2+^ influx and increases BDNF release. BDNF activates the tropomyosin receptor kinase B (TrkB), leading to the downregulation of the K^+^-Cl^−^ cotransporter KCC2 and the upregulation of the N-methyl-D-aspartate (NMDA) receptor, both of which contribute to the occurrence of pain. Increased Ca^2+^ influx enhances the release of the excitatory neurotransmitter glutamate, which activates NMDA and α-amino-3-hydroxy-5-methyl-4-isoxazolepropionic acid (AMPA) receptors, thereby mitigating hyperalgesia and hypersensitivity. Furthermore, NGF from injured SCs activates the TrkA receptors, leading to the upregulation of substance P (SP), calcitonin gene-related peptide (CGRP), and the transient receptor potential cation channel subfamily V member 1 (TRPV1) receptors, also known as capsaicin receptors. SP triggers the neurokinin 1 (NK1) receptor, and CGRP reduces the degradation of SP, leading to pain sensation. NGF can also increase the expression of BDNF, thereby affecting pain perception.

Recent results have demonstrated that LRP1 in SCs may control RAG expression by sensory neurons, which may be linked to chronic pain [468,469]. This supports the notion that SCs may be an essential target for blocking pain following peripheral nerve damage.

SCs can also relieve neuropathic pain by remyelinating injured nerves [19]. Therefore, therapeutic targeting or transplantation of SCs is of great interest and importance in designing novel therapeutic approaches to neuropathic pain. Transplantation of microencapsulated SCs next to the injured sciatic nerve in the rat model of chronic constriction injury significantly improved the healing of myelin sheaths in the injury areas compared to those without SCs [470]. Notably, SC administration decreased the expression of ATP receptors P2X2 and P2X3 that transmit algesia and nociception signals by sensory neurons.

Inhibition of SC pannexin 1 attenuates neuropathic pain by suppressing inflammatory responses [471]. Conversely, genetically targeting Grin1, which encodes the essential GluN1 NMDA receptor subunit in SCs, prompts hypersensitivity in pain processing in the absence of nerve injury [472].

Another vital aspect of SC regulation of pain sensation is associated with cancer pain, often resulting from the damage to the surrounding organs and erosion of neural tissues during tumor growth, invasion, metastasis, or from cancer treatment. Almost 50% of patients with cancer, especially at advanced stages, suffer from chronic pain, which is often persistent and intense [473]. Accumulating evidence suggests the role of SCs in cancer pain via, for instance, increased secretion of pro-nociceptive mediators such as TNF-α, NGF, glutamate, BDNF, TLR ligands, and IL-6 [474,475,476]. Although the use of SCs in cancer pain therapy is in its early stages of development, numerous studies have reported the therapeutic effects of SCs on neuropathic pain [476]. For instance, transient receptor potential cation channel subfamily V member 4 (TRPV4) activation in SCs has been shown to mediate mechanically induced pain in both in vitro and in vivo mouse cancer models. Conversely, TRPV4 inhibition decreased mechanical nociception in tumor-bearing mice in a dose-dependent manner [477]. Moreover, selective in vivo SC transduction and functional block of Piezo1 channel activity in SCs using an AAV vector have been shown to alleviate mechanical hypersensitivity following nerve injury in rats [478]. Therefore, SC-based approaches might initiate new directions in cancer pain relief.

## 5. Schwann Cells as a Therapeutic Target for Peripheral Neuropathies

Many PNS diseases affect the SC function and differentiation phenotype (Figure 1). With recent progress in understanding the mechanisms of crosstalk between SCs and peripheral neurons, it is clear that SCs play a central role in the pathogenesis of various inherited, metabolic, and inflammatory neuropathies [9,288,313]. The best example is the well-characterized role of SCs in the pathogenesis of CMT disease, diabetic neuropathy, and neuropathic pain. The progression of neurologic autoimmune diseases, such as GBS, which damages the myelin sheath of the PNS, ultimately affects the function of SCs, complicating the primary pathogenic mechanisms [352].

A growing body of evidence suggests that targeting abnormal SCs in various types of demyelinating neuropathies may be clinically beneficial (Figure 2). Historically, neurotransmitters, neurosteroids, and neurohormones have presented valuable opportunities for investigating how pharmacological interventions targeting SCs can protect or stimulate the recovery of regenerative functions in SCs in peripheral nerve disorders [313,479]. For example, inhibition of neuronal nitric oxide synthase or activation of heme oxygenase, which regulates oxidative stress in SCs, may protect against peripheral nerve degeneration [480,481]. Minocycline, a broad-spectrum antimicrobial tetracycline, has been shown to protect SCs from ischemia-like injury [482]. Interestingly, lithium represents a fascinating pharmacological agent, unique in its ability to block the onset of myelination without promoting myelin degradation and SC dedifferentiation, while maintaining the integrity of pre-existing myelinated fibers [483]. Roflumilast, a selective phosphodiesterase-4 inhibitor, which is commonly used to treat chronic obstructive pulmonary disease, plaque psoriasis, and atopic dermatitis, has been reported to promote the differentiation of SC to a myelinating phenotype associated with increased expression of myelin proteins, such as MBP and MAG [484]. Treated SCs improved axonal outgrowth, which was accompanied by faster myelination.

Furthermore, recent studies have demonstrated that antineoplastic drugs, such as Epothilone B, a chemotherapeutic microtubule inhibitor, inhibit the PI3K/Akt signaling pathway, thereby stimulating SC motility without affecting SC survival and thus promoting axonal regeneration after peripheral nerve injury [485]. Deferoxamine, an iron and aluminum chelator used for the treatment of various diseases, including acute iron poisoning and hemochromatosis, could stimulate SC viability, proliferation, and migration and upregulate the expression of myelin-related and nerve growth-promoting genes in SCs, therefore promoting peripheral nerve regeneration [486].

Multiple data also suggest that new approaches should be developed to expand the repair and supportive functions of SCs and avoid their deterioration. Focusing on signaling pathways, transcription factors, epigenetic mechanisms, and microRNA networks in SCs under various pathophysiological conditions is highly justified to elucidate the mechanisms of selectively regulating SC injury response, plasticity, and recovery potential for therapeutic purposes [30,487]. For instance, new data highlight the importance of epigenetic control of gene expression in normal and repair SCs, which regulate cell differentiation and myelinating function [488,489]. The acetylation of histones, mediated by histone acetylases and histone deacetylases (HDACs), is a key epigenetic pathway, and HDAC inhibitors can improve nerve regeneration outcomes [490]. Furthermore, epigenetic regulation by HDACs in SCs after injury accelerates the de- and redifferentiation pathways in SCs, supporting several stages of nerve repair, at least partially, by controlling the expression of inflammatory cytokines [491]. The HDAC6 inhibitor, CKD-504, regulates chaperone heat shock proteins HSP90 and HSP70, which are involved in the folding/refolding of PMP22. CKD-504 treatment reestablished myelination in both MSC-derived SCs from CMT1A patients and in the sciatic nerves of C22 mice, thereby amending the axonal integrity of the sciatic nerve and resulting in behavioral, electrophysiological, and histological progress in C22 mice [492]. Thus, targeting epigenetic regulation of resident repair SCs by HDAC inhibitors after peripheral nerve injury may represent an interesting and promising approach. In fact, new data show that the ablation of HDAC8 in SCs accelerates the regrowth of sensory axons and the recovery of sensory function, through the control of the E3 ubiquitin ligase TRAF7, destabilization of hypoxia-inducible factor 1-alpha (HIF1α), and phosphorylation of c-Jun in SCs ensheathing sensory axons [493].

Multiple signaling pathways in SCs may serve as potential targets. Low-density lipoprotein receptor-related protein 4 (LRP4) is expressed in SCs and may control peripheral nerve regeneration, as conditional knockout of LRP4 accelerated demyelination and enhanced the proliferation of SCs in injured nerves, probably by downregulating the Krox-20 and MPZ pathways [494]. This indicates that LRP4 in SCs may be a potential therapeutic target for peripheral nerve recovery. Carvacrol, a primary component of the herb *Origanum vulgare*, can successfully protect against experimental peripheral nerve degeneration by inhibiting the upregulation of transient receptor potential melastatin M7 (TRPM7) in SCs, suggesting its potential pharmacological prophylactic application [495]. Many additional examples are presented in other sections discussing SCs in specific pathological conditions.

Targeting the immunomodulatory function of SCs [327,328,496] may serve as another interesting therapeutic approach. For instance, myeloid-derived suppressor cells (MDSCs), a mixed population of regulatory immune cells, may help resolve inflammatory responses and enhance functional recovery in models of injury and inflammation-mediated degeneration [497]. Given recent data demonstrating that activated repair-like SCs can attract and upregulate the immunosuppressive activity of MDSCs [498] and control the immunosuppressive and exhausting phenotype of T cells [499], targeting SCs in inflammatory neuropathies may serve as an alternative therapeutic approach.

## 6. Gene Therapy to Target and Modify Schwann Cells

Due to their ability to support and control axonal regeneration and remyelination following nerve injury and certain types of demyelinating neuropathies, SCs are the essential clinical target in related diseases and medical conditions (Figure 2 and Figure 3). Identifying causative genes or elucidating the molecular mechanisms of SC dysfunction in certain neuropathies has provided the basis for designing gene-modifying therapies. Novel developments in gene therapy methodologies have permitted the transfer of a functional replica of a disease-causing gene and the correction, replacement, or silencing of disease-causing genes in gene-targeting strategies [500].

### 6.1. Gene Therapeutic Targeting of Schwann Cells in CMT Disease

Gene therapy approaches, such as gene silencing or gene replacement strategies, targeting SCs have been relatively well evaluated in CMT disease and CMT animal models, demonstrating promise. This is because many genes involved in demyelinating forms of CMT disease are expressed primarily or exclusively by myelinating SCs. Additionally, many knockout mouse models of different CMT forms recapitulate the main features of CMT disease and provide an appropriate model to evaluate targeted genetic therapies to SCs [501].

Sargiannidou et al. and Kagiava et al. were probably the first to report that a single into-sciatic nerve or intrathecal injection of a lentiviral vector with a myelin-specific promoter resulted in targeted expression of connexin 32 in adult myelinating SCs throughout the PNS in *Gjb1*^−/−^ mice, a genetically authentic model of CMT1X. They improved the nerve pathology [502,503]. Many studies confirmed the therapeutic potential of this gene addition. *Gjb1*^−/−^ mice treated with lentiviral vectors encoding the *GJB1* gene after the onset of peripheral neuropathy showed improved motor performance at 10 months [504]. Intrathecal injection of lentiviral vectors encoding the *GJB1* gene in transgenic Cx32 knockout mice harboring interfering T55I, R75W, or N175D CMT1X mutations to reflect different mutations in human CMT disease showed different levels of improvement in motor performance and fiber myelination [505]. In another study, *Sh3tc2*^−/−^ mice, representing CMT disease type 4C, were treated with an intrathecal injection of a lentiviral vector to drive the expression of the human SH3TC2 cDNA under the control of the Mpz promoter specifically in myelinating SCs [506]. Gene replacement therapy significantly improved motor performance and nodal molecular architecture and reduced blood neurofilament light chain levels, a marker of neuroaxonal injury. In the same animal model of CMT4C, the utilization of an AAV9 vector and the MPZ promoter to achieve SC-targeted expression of SH3TC2 also demonstrated treatment improvements in motor performance tests, along with a decreased ratio of demyelinated fibers and elevated myelin thickness [507]. Lee et al. used a lentivirus encoding miR-381 to reduce the expression of PMP22 in C22 mice, a CMT1A mouse model injected intraneurally distal to the sciatic notch [508]. Treated mice demonstrated increased motor nerve conduction velocity and improved structural abnormalities in the myelination of the sciatic nerves.

Thus far, parallel studies have determined the feasibility of using AAV vectors to target SCs in CMT disease [509]. A single lumbar intrathecal injection of the AAV-based gene construct in knockout CMT mice for targeted expression of connexin32 in SCs resulted in extensive gap junction protein biodistribution in the PNS, improved myelination and motor performance, and decreased inflammation in peripheral nerves [510]. Similar data on SC-targeted delivery of viral vectors encoding specific proteins confirmed the previous proof of principle for a clinically adaptable gene therapy approach to treat CMT, both before and after the onset of neuropathy [511,512].

Silencing mutated or overexpressed genes is another encouraging approach to amend the phenotype of certain forms of CMT [442]. Indeed, small interfering RNA (siRNA), AAV-delivered small hairpin inhibitory RNA (shRNA), and AAV-based microRNA (miRNA) have been tested to downregulate PMP22 mRNA expression in SCs in various CMT1A model systems [513]. Intra-nerve administration of an AAV vector expressing shRNAs targeting PMP22 in the sciatic nerve resulted in transgene expression in resident myelinating SCs in mice, rats, and non-human primates [514]. This gene therapy upregulated myelination and prevented motor and sensory losses for over 12 months in a rat model of CMT1A. AAV vector-based delivery of miR871 by lumbar intrathecal injection into C61-het mice, another CMT1A model, effectively transduced SCs in peripheral nerves and reduced PMP22 mRNA and protein expression [515]. This was associated with markedly improved functional outcomes, nerve conduction velocities, and ameliorated myelin pathology in the lumbar roots and femoral motor nerves.

Interestingly, new data showed that non-viral delivery of siRNA that specifically and selectively reduced the expression level of the *PMP22* mutant allele alleviated the demyelinating neuropathic phenotypes of CMT in Tr-J mice, which was associated with the reversion of the low viability of SCs, improved motor function and muscle volume, and amplified expression of myelinating proteins MBP and MPZ [516]. Similarly, the administration of siRNA PMP22 conjugated to squalene nanoparticles caused normalization of PMP22 protein levels, regeneration of myelinated axons and myelin compaction, and restored locomotor activity in two transgenic CMT1A mouse models [517]. Furthermore, antisense oligonucleotides that bound to the 3′-untranslated region of *PMP22* could restore myelination and motor nerve function in two murine CMT1A models [518].

### 6.2. Genetic Engineering of Schwann Cells for Nerve Injury Treatment

Genetic modification of SCs can control the expression of specific factors required for peripheral nerve regeneration (Figure 2). For instance, the knockdown of histone deacetylases in SCs abrogated the expression of the transcriptional regulators of myelination SOX10 and Krox20 [488]. Transfection of SCs to overexpress NGF, GDNF, or the transcription factor c-Jun promoted not only the upregulated expression of neurotrophic factors, which in turn led to neurite outgrowth through both autocrine and paracrine mechanisms, but also improved regenerative cellular processes [519]. GDNF-expressing SCs enhanced the formation of myelin sheath and nerve conductance [83]. Transplantation with SCs overexpressing nuclear factor NRF2 increased vascularity and nerve density in crushed sciatic nerves of rats, which was associated with improved survival of SCs [298]. SCs transduced to overexpress fibroblast growth factor (FGF) demonstrated an enhanced ability to promote neuron regeneration after sciatic nerve injury [520]. In a similar model, implanted SCs overexpressing NT-3 prevented neuron death, increased axon regeneration, and improved the damaged nerve function [521].

Transfection of SCs with a c-Jun-encoding lentiviral vector resulted in the upregulated expression of various growth factors, including GDNF, NGF, BDNF, and artemin [522]. It significantly increased the ability of SCs administered with poly (ε-caprolactone) nerve conduits to improve the survival and elongation of injured sensory neurons and the sciatic function index, and alleviate target muscle atrophy and muscle action potential [523].

SC targeting via intrasciatic injection of adeno-associated vectors is another promising gene therapy strategy for peripheral nerve regeneration. AAV2/8 preferentially transduced SCs in vivo, and injection of AAV2/8 encoding the ciliary neurotrophic factor (CNTF) resulted in upregulated expression of myelin proteins P0 and PMP22 after transduction of injured sciatic nerves. This was associated with significantly improved axonal regeneration and the compound muscle action potential [524]. New findings demonstrated that mechanosensitive ion channels Piezo1 and Piezo2 are highly expressed in SCs [525,526]. Genetic ablation experiments revealed that Piezo1 is an inhibitor of radial and longitudinal myelination in SCs, while Piezo2 is required for myelin formation [526]. Selective in vivo SC transduction via intrasciatic delivery of AAV encoding shRNA against Piezo1 diminished the development of mechanical hypersensitivity after nerve injury [478].

Interestingly, it was reported that elevated long non-coding RNA (lncRNA) metastasis-associated lung adenocarcinoma transcript 1 (MALAT1) expression in an injured sciatic nerve promoted the proliferation and migration of SCs by sponging miR-129-5p and increasing BDNF expression and secretion [527]. Sciatic nerve injury may also result in the upregulation of long non-coding RNA (lncRNA) axon regeneration-associated transcript (lncARAT) in SCs and SC exosomes, contributing to axonal regeneration. LncARAT absorbs miRNA-329-5p in macrophages, causing STAT-1/6-dependent elevated expression of cytokine-inducible regulator, suppressor of cytokine signaling (SOCS-2). This, in turn, induces pro-regenerative activity in macrophages [528]. Thus, lncARAT in SCs may be a promising therapeutic tool for peripheral nerve repair. In contrast, miR-328a-3p, which is also upregulated after nerve injury, inhibits SC proliferation, migration, and viability [529]. Lowering miR-328a-3p levels, for instance, by specific lncRNA, might support SC function after nerve damage and stimulate nerve regeneration. Remarkably, it has recently been reported that reducing miR-146a-5p in SCs and subsequently in SC exosomes leads to type 1 macrophage polarization via the TRAF6/NF-κB pathway, affecting peripheral nerve injury [530].

These and similar findings open new opportunities for utilizing genetic engineering of SCs to treat peripheral nerve damage.

### 6.3. Gene Therapeutic Targeting of Tumorigenic Schwann Cells

The additional potential object for targeted gene therapy is the tumorigenic SCs that trigger neurofibromatosis and schwannomatosis, as discussed below. For instance, high efficacy in transducing primary human SCs isolated from a plexiform neurofibroma using selected recombinant AAV vector variants has been recently described [500]. Furthermore, numerous approaches for gene-targeted therapies have been introduced and tested in models related to neurofibromatosis and schwannomatosis, opening the path to clinical trials [531].

Schwannomas are also tempting targets for gene therapy due to their slow growth rate and legibility for localization. For neurofibromatosis type 1 and schwannomatosis, gene-targeted therapies are commonly intended to upregulate the levels of functional proteins in cells that express either one or no functional copies of a tumor suppressor gene [531]. Gene therapy tactics for NF2-related schwannomas comprise a suicide gene, gene replacement, and combination gene knockdown and replacement approaches [532]. The treatment of nude mice bearing human and mouse schwannomas with an oncolytic replication-conditional herpes simplex virus vector induced tumor shrinkage or complete tumor regression [533]. In 2010, HSV-1 amplicon vector-based intratumoral delivery of caspase-1 under the SC promoter was reported to cause regression of schwannoma in a xenograft mouse model [534]. Further published results showed that utilizing the P0 promoter, selectively expressed in the SC lineage, effectively avoids transgene expression in non-Schwann-lineage cells, such as neurons, and prevents any neural toxicity [535]. Injection of an AAV vector, encoding the pro-apoptotic gene (caspase-1) under the P0 promoter in nude mice with the sciatic nerve implanted immortalized human schwannoma cells, induced tumor regression. Even more, the same gene therapy caused tumor regression and prevented tumor-associated pain in another NF2 xenograft model [535]. An AAV1-based vector encoding gasdermin, a substrate of caspases and an effector molecule for pyroptosis, under the P0 promoter also demonstrated high antitumor efficacy in the same tumor model [536]. Bai et al. also confirmed the feasibility of gene replacement therapy for NF1-related tumors by achieving beneficial transduction efficacies and functionality of AAV-based transduction of SCs to restore NF1-affected signaling pathways [537].

Ahmed et al. studied the apoptosis-associated speck-like protein containing a caspase recruitment domain (ASC) as a putative schwannoma tumor suppressor and reported that ASC expression was downregulated in 80% of the tested human schwannoma cells [538]. ASC transfection of tumor cells induced schwannoma cell death associated with triggered caspase activation. AAV-mediated intratumor gene therapy in a human xenograft schwannoma model suppressed tumor growth by increasing tumor cell apoptosis and fixed tumor-associated pain without noticeable toxicity. The effectiveness of this schwannoma gene therapy approach was proven in a murine schwannoma model, suggesting that it may be translated to human clinical trials [536]. A gene replacement strategy via intratumoral administration of an AAV vector encoding merlin in human NF2-null SC-derived tumors in the sciatic nerve of nude mice also demonstrated high efficacy, inducing tumor regression associated with tumor cell apoptosis [539].

Knockdown and overexpression experiments with SC lines, including NF1-associated malignant peripheral nerve sheath tumor cells, NF1-heterozygous deficient SCs, and NF1-deficient plexiform neurofibroma cells, to target Forkhead box M1 (FOXM1), a classical proliferation-associated transcription factor, showed that FOXM1 could mediate human malignant peripheral nerve sheath tumor development and could be a promising therapeutic target for tumor treatment [540].

Together, these data successfully prove the concept of SC targeting for gene silencing and gene replacement and suggest that these tools could benefit different neuropathological conditions and disorders involving SC functionality [531,532,541].

## 7. Schwann Cell-Derived Factors for Therapeutic Use

SC-secreted molecules maintain axonal integrity and promote regeneration, while also controlling various cells in the neuronal microenvironment by regulating physiological processes such as cell differentiation, proliferation, homing, and gene expression. Protein expression profiling of primary human SCs identified almost 20,000 unique peptides corresponding to more than 1550 individual proteins [542]. SC-secreted signals, collectively referred to as the secretome, include, but are not limited to, growth and neurotrophic factors, cytokines, exosomes, and ECM proteins and factors. These are currently being tested from an application viewpoint as SC-derived biomaterials for nerve injury repair, axonal remyelination, and neuroprotection.

The SC secretome plays a crucial role in orchestrating Wallerian degeneration and promoting axonal regeneration [543]. Indeed, the diminished ability of SCs to sustain the secretion of neurotrophic factors over a prolonged period may explain the collapse of regeneration in continually denervated nerves. Furthermore, SCs are known to produce neurosteroids that can, directly and indirectly, control myelin formation via binding to the classical steroid receptor in SCs and neurons [544,545,546].

### 7.1. Schwann Cell Extracellular Vesicles and microRNA for Therapeutic Applications

Extracellular vesicles (EVs) are a common term for several types of small-diameter vesicles produced and secreted by different cells that contain numerous molecules such as proteins, lipids, and nucleic acids (DNA, microRNA, circRNA, lncRNA) [547]. EV therapy offers advantages over cell-based therapy, such as ease of engineering and a lower risk of tumorigenesis [548]. Consequently, as a cell replacement therapy, isolated EVs and related engineering derivatives have been developed and applied to the diagnosis and treatment of different pathological conditions, including neurological diseases [548,549,550,551].

Exosomes are EVs with diameters ranging from 30 to 150 nm, derived from the endoplasmic reticulum. They play a fundamental role in cellular communication and are promising vehicles for cell-specific molecule and drug delivery [552]. SC exosomes are important for axonal protection and regeneration following nerve injury. The exosomes produced by repaired SCs, unlike exosomes from differentiated SCs, upregulate the expression of miRNA-21, c-Jun, and Sox2 in SCs, which stimulate differentiation and migration of SCs and support axonal regeneration [553]. EVs secreted by SCs may also encourage axonal elongation and regeneration of neurons via the Akt/mTOR/p70S6K pathway [554]. SC EVs have been reported to promote axonal regeneration in vitro and advance regeneration after sciatic nerve damage in vivo [552,555]. The potential therapeutic value of SC-derived exosomal vesicles has been highlighted by promising engineering strategies to customize SC exosomes for specific activities and demonstrations of how SC exosomes provide exceptional clinical advantages over SC transplantation for repairing the wounded spinal cord [556,557].

SC exosomes have been reported to improve functional recovery after spinal cord injury. For instance, Pan et al. found that SC exosomes promote axonal protection and functional recovery after spinal cord injury by upregulating autophagy, reducing apoptosis via the EGFR/Akt/mTOR signaling pathway, augmenting Toll-like receptor 2 expression in astrocytes, and decreasing the deposition of chondroitin sulfate proteoglycan [558,559]. Newer data from that team revealed that milk fat globule-epidermal growth factor-factor 8 (MFG-E8) within SC exosomes is responsible for suppressing M1 polarization and stimulating M2 polarization of macrophages via the SOCS3/STAT3 signaling pathway, thereby improving the inflammatory microenvironment and inhibiting neuronal apoptosis [560]. In vitro and in vivo experiments have also demonstrated that SC exosomes can be taken up by vascular endothelial cells and promote functional recovery after spinal cord injury by enhancing angiogenesis through the delivery of integrin-β1 [561]. SC-derived EVs have been shown to significantly improve motor and sensory function recovery in rats with long-distance sciatic nerve defects treated with chitosan nerve conduits [562]. The ability of SC EVs to improve regeneration of the damaged sciatic nerve in vivo and promote neurite outgrowth of sensory and motor neurons in vitro has been demonstrated by multiple studies [554,563,564,565,566].

Recently, Nishimura et al. reported that the systemic administration of human SC exosomes significantly decreased the overall contusion volume, microglial activation, secondary inflammatory injury, and histopathological damage in rats with traumatic brain injury [567]. A new study has confirmed multilayered therapeutic advantages of human SC-derived exosomes, targeting both acute and chronic neuroinflammatory pathways to foster functional recovery.

Jugular infusion of SC EVs after traumatic brain injury reduced the acute pro-inflammatory reaction in the ipsilateral cortex and hippocampus, as assessed by a decrease in inflammasomes and inhibition of the STAT3/pSTAT3/SOCS3 pathway. Reduction of cerebral edema and contusion volumes confirmed the diminishing of subacute histopathological changes and a decrease in microglial activation. This was associated with the preservation of both sensorimotor function and cognitive performance. Moreover, a decline in circulating neutrophils indicated a reduction in chronic systemic inflammation [568].

SC-derived exosomes have also been reported to improve peripheral neuropathies. Normal exogenous SC-derived exosomes have been shown to ameliorate peripheral neuropathy in mice induced by abnormal microRNA biogenesis in resident SCs [569]. MicroRNAs are non-coding RNAs that regulate post-transcriptional protein expression and affect nearly 60% of mammalian genes [570]. Wang et al. investigated the therapeutic potential of exosomes derived from healthy SCs in treating diabetic peripheral neuropathy. They reported that the intravenous injection of SC EVs in Type 2 diabetic *db*/*db* mice with peripheral neuropathy significantly improved the sciatic nerve conduction velocity and thermal and mechanical sensitivity [571]. The corrected neuropathy was associated with the extension of epidermal nerve fibers, remyelination of sciatic nerves, and reversion of diabetes-reduced mature form of miRNAs 21, 27a, and 146a in sciatic nerve tissues. In vitro data revealed that SC exosomes stimulated neurite outgrowth in diabetic DRG neurons and altered the migration of SCs in response to high glucose [571]. New results showed functional impairments in SCs cultured from the sural nerve biopsy of an ALS patient. But their treatment with exosomes isolated from cadaver donor SCs markedly recovered their growth potential in vitro [197].

Interestingly, mechanical stimulation of SCs may alter the microRNA composition of their nanoscale vesicles by upregulating miR-23b-3p, which reduces neuronal neuropilin 1 expression responsible for neurite outgrowth in vitro and nerve regeneration in vivo [274]. Additionally, neuron-to-glia communication may regulate the content of SC-derived pro-regenerative exosomes and their transport to axons, thereby modifying axonal elongation. ATP release from activated sensory neurons has been shown to stimulate P2Y receptors in repair SCs, thereby increasing the level of miRNA-21 present in SC EVs [572].

Likewise, SC-derived exosomes may restore paclitaxel-induced peripheral neuropathy. The results of in vitro experiments demonstrated that SC EVs protected the DRG neurons from paclitaxel damage, while in vivo data revealed that SC exosomes ameliorated paclitaxel-induced plantar intraepidermal nerve fiber loss, DRG injury, and mechanical nociceptive sensitization in rats [573]. The mechanistic experiments uncovered that SC exosomes facilitated axonal regeneration and protected injured neurons by increasing miR-21, which suppresses the phosphatase and tensin homolog (PTEN) signaling pathway, recovering paclitaxel-induced peripheral neuropathy.

Thus, given its versatile contribution to angiogenesis, neurogenesis, axon remodeling, and anti-inflammatory response, SC exosomes may be a promising next-generation therapeutic modality for improving neurorestorative treatments. Notably, a recent study has established a feasible framework for the isolation and thorough omics characterization of SC EVs, which, importantly, is consistent with their therapeutic properties in neurological applications [574].

### 7.2. Schwann Cell-Derived Extracellular Matrix Proteins and Factors

During the development and regeneration processes, SCs secrete extracellular matrix factors containing a diversity of macromolecules, such as collagens, laminins, proteoglycans, elastin, fibronectin, entactin, etc. Remarkably, in the PNS, fibronectin is primarily expressed by SCs and located mainly within the perineurium [238]. ECM components produced and secreted by SCs are deposited beside SCs to support the basal lamina sheets surrounding individual axon-SC units. SC ECM thus regulates axonal growth during development and regeneration [575]. In addition, both the secretion and assembly of ECM components by SCs control SC polarization and are required to develop the key SC functions—ensheathment and myelination [576]. Surprisingly, SCs can regulate their ECM microenvironment to safeguard directional neuron migration. It has been shown that sensory neurons and motoneurons require the SC-derived Sox2-dependent fibronectin matrix to migrate along the oriented SCs [577]. Even more importantly, Sox2 and fibronectin are co-expressed in pro-regenerative SCs in vivo during sciatic nerve regeneration in a time-dependent manner.

As SC transplantation is considered the gold standard in cell-based treatment for peripheral nerve injuries, ECM molecules from SCs are one of the most appropriate biomaterials for nerve repair [236,238]. Such cell-free transplantation methods, along with their combinations, offer excellent promise for clinical translation.

Gu et al. developed a chitosan/silk fibroin-based, SC-derived, ECM-modified scaffold suitable for peripheral nerve tissue engineering applications [578]. Its regenerative properties were verified when it was used to bridge a 10 mm gap in the rat sciatic nerve model. Additionally, blood and histopathological parameters confirmed the safety of scaffold modification by SC-derived ECM. Other studies have confirmed these promising results [579,580]. Interestingly, ascorbic acid can stimulate the secretion of ECM by SCs, and the myelinating ability of SCs expressing collagen I, collagen IV, fibronectin, and laminin has also been demonstrated [581].

ECM substantially impacts the behavior and activity of resident cells, including SCs, maintaining the physiological functions of tissues and organs and their healing potential. Diverse components extracted from the ECM material of peripheral nerves, which are usually bioactive, biocompatible, and well tolerated in vivo, can induce SC proliferation and alignment, thus serving as an alternative to SCs as cell-free scaffolds for nerve regeneration [239]. Importantly, ECM proteins modulate SC morphology, activity, and protein expression and may accelerate SC conversion to a regenerative phenotype [582]. Several studies have reported that compositional decellularized nerve ECM initiates SC alignment and nerve fiber remyelination in vitro and in vivo [583,584,585,586]. Different ECM scaffolds extracted from PNS materials provide suitable platforms for SC transplantation, offering a feasible alternative to cell-free nerve grafts [239].

Thus, further studies addressing the interaction between ECM cues and SCs, including SC-derived ECM constituents, may offer novel cell-based and cell-free approaches to augment PNS regeneration and improve neuromodulation treatments.

## 8. Schwann Cells in Wound Healing

Wound healing is a complex physiological process that requires synchronized and orchestrated communication between numerous cell types with distinct functions to restore the original homeostatic conditions. Substantial experimental and clinical data suggest that innervation plays a key role in wound healing, tissue regeneration, and organ repair pathways [587]. For instance, the absence of innervation is known to impair skin healing, and individuals with peripheral neuropathy often exhibit a failure to heal [588,589]. The skin is a tissue densely innervated by sympathetic autonomic and primary afferent sensory nerve fibers providing pain, pruritus, thermal, and tactile sensations [590]. The skin nerve fibers are accompanied by myelinating and non-myelinating SCs, which, in addition to their well-known function, may also provide nociceptive and touch sensation [7,10,591].

Neuromediators, including neuropeptides, neurohormones, cytokines, and growth factors, are released by immune cells, keratinocytes, and nerve terminals in response to skin lesions and peripheral nerve damage, and are also involved in various stages of wound healing [592]. While the dissemination of SCs from the disrupted PNS nerves into the granulation tissue within the wounded skin, their proliferated and dedifferentiated, have been well studied (see above), recent data demonstrated that genetic ablation of SCs significantly deferred wound contraction and closure, reduced development of myofibroblasts, and diminished re-epithelization in the repairing skin [593]. Similarly, tempered injury responses and delayed SC dedifferentiation have been reported to impair skin wound healing [594]. It has been established that early activation of SCs after injury creates a permissive microenvironment, while the later redifferentiation of SCs ensures the formation of a myelin sheath. This modulation of the SC phenotype is controlled by c-Jun and Sox2—regulators of SC reprogramming, and Krox-20 and Sox10—regulators of SC myelination [595].

Recent analysis of SC activity using a cell atlas of the wound healing process, generated by integrating single-cell RNA sequencing, revealed that the Wnt signaling pathway regulates SC dedifferentiation at an early stage of wound repair. Wnt inhibition blocked SC dedifferentiation, resulting in defective repair [596]. Furthermore, SC-secreted TGF-β endorsed the motility of keratinocytes and fibroblasts, and TGF-β improved the healing of chronic wounds with impaired SCs [596]. Injury-activated SCs regulate the formation of myofibroblasts via paracrine modulation of TGF-β signaling [593]. Similarly, analysis of skin repair pathways in other models has demonstrated that suspended wound healing in diabetes may be due to a weakened SC repair reaction and diminished effects on myofibroblast differentiation [594]. In addition, a subset of specialized profibrotic SCs with reprogrammed genetic activity may promote the formation of abnormal scars, such as hypertrophic scars or keloids, during the nerve repair process in the injured skin [597]. These SCs were not associated with axons, displayed a nonclassical repair-like phenotype, and supported the overproduction of the extracellular matrix. Therefore, the authors hypothesized the existence of two types of repair/repair-like SCs: Type 1, which appears temporarily after acute injury, and Type 2 with profibrotic properties, which may persist for a prolonged time after tissue regeneration [598]. New data demonstrate that SCs in keloid tissue of the rabbit ear scar model significantly upregulate the expression of the insulin-like growth factor binding protein 5 (IGFBP5) gene, which is positively associated with keloid fibroblast proliferation, migration, invasion, angiogenesis, and cell cycle progression [599]. It is vital to note that identifying the specific set of genes in so-called keloidal SCs should help understand the contribution of SCs to the pathogenesis of other fibrotic disorders, which have devastating consequences for various organs, including the heart, lungs, liver, kidneys, and skin.

These data discovered a novel role for cutaneous SCs. They demonstrated that SCs could be used as a therapeutic target or treatment tool to improve dermal wound healing and reduce scar formation [27]. Collectively, numerous findings from other experimental wound healing and regeneration models demonstrate the pivotal role of nerve-associated SCs in blastema-mediated regeneration, digit regeneration, wound healing, and organ regeneration, as SCs provide trophic factor support, a source of progenitor cells, nerve repair coordination, and assist skin wound closure [587,600,601]. SCs can also promote vascular regeneration owing to the expression of various pro-angiogenic factors.

SCs are also known to contribute to bone regeneration. In the alveolar bone injury model with tooth extraction, dedifferentiated repair SCs either formed new axons or transdifferentiated into osteoblasts during the healing process [29]. Scaffold vascularization and osteogenesis in a femur defect model induced by bone grafts and endothelial cells was significantly accelerated after the addition of SCs [602]. Similar effects on angiogenesis and osteogenesis can be achieved by utilizing SC-derived exosomes [603]. SC exosomes have been reported to upregulate the proliferation, osteogenic, and neurogenic differentiation of human periodontal ligament cells, as well as the expression of angiogenic factors in vascular endothelial cells [599]. Moreover, in a rat model of periodontal bone defects, SC exosomes accelerated the attraction of endogenous cells, controlled neural and vascular neogenesis, and facilitated periodontal bone regeneration [599]. Importantly, accelerated formation of bones by SC-derived exosomes in vivo was dose-dependent [604]. The elimination of SCs by denervation significantly delays the healing process in various models [605,606]. Interestingly, new research has revealed that dental pulp stem cells can reduce pyroptosis and mitochondrial ROS production in SCs through mitochondrial transfer, thereby enhancing nerve regeneration [607]. A comprehensive understanding of SC involvement in regenerative processing across different tissues should provide a better perspective for integrating these pathways into a therapeutic framework to promote tissue and organ repair.

Finally, one unusual aspect of SCs’ reparative function is their potential association with a significant clinical phenomenon: the formation of surgery-associated metastasis. The fact that the surgical insult might trigger or accelerate locoregional and distant tumor recurrence has been acknowledged for a long time, and more recent evidence demonstrated that the surgical operation or even tissue biopsy may generate a permissive environment for tumor growth and the appearance of metastases [608,609,610,611,612,613]. Although the hypothesis that primary tumors can inhibit the growth of metastases—“concomitant resistance” and “cancer hormesis”—is still under discussion [614,615,616,617,618], numerous experimental studies provide more mechanistical results. The mechanisms of surgery-induced metastatic processing involve the promotion of wound healing and functional recovery connected with the stimulation of immunosuppressive, inflammatory, and pro-angiogenic pathways, as well as the promotion of single circulating malignant cells and the disbalance of dormant niches and escape from dormancy [619,620,621,622].

Furthermore, it is well established that nerve axons and SCs are present in human and animal tumors and are capable of accelerating tumor growth and progression (see below). As discussed above, a traumatic injury to neuronal axons within the resected tumor and the surrounding tissue initiates Wallerian degeneration and nerve injury-induced “repair” SCs. In addition, tumor-associated SCs exhibit a repair-like or injury response-like phenotype, as described below. It was also speculated that glial cell failure might happen if adaptive strategies established by tumor-activated SCs collapsed or if the program of SC-driven tissue repair terminated in a maladaptive response. This can then promote or support the metastatic disease [613]. Tumor-associated “repair-like” SCs encourage metastasis in surgery- or biopsy-related wound healing even more strongly because surgical stress and wound healing signaling contribute to the higher degree of SC activation, proliferation, motility, and dedifferentiation in the regenerating tumor microenvironment [613]. However, the role of SCs in metastasis development as an outcome of wound healing has never been experimentally proven. Further studies are needed, as the results should provide a new target for safe and expedited wound healing and tissue regeneration following cancer-associated surgical procedures.

## 9. Schwann Cells in Cancer

The clinical aspects of tumor innervation and the role of the PNS in the development, growth, and progression of solid tumors have been investigated for decades [623]. While the direct and indirect modulation of tumor growth and metastasis by the autonomic and sensory neurotransmitters and neuropeptides has been proven in numerous preclinical and clinical studies [624,625,626,627], the involvement of SCs in the formation of the tumor neuroenvironment and maintenance of the neuro–immune axis in cancer has been documented only recently [498,628,629,630]. These studies have revealed the functional role of SCs in promoting the proliferation and motility of neoplastic cells, modulating antitumor immune responses, and conveying pain sensations in cancer. Altogether, this suggests that SCs are legitimate targets for anticancer therapy and could also be used as a therapeutic tool for treating at least certain types of malignancies.

### 9.1. Schwann Cells in Tumorigenesis

The regenerative, immunomodulating, and pro-angiogenic properties of SCs, as well as their functional and phenotypic plasticity, make SCs an exceptional target for malignant cells to exploit and convert into supporters of carcinogenesis and metastasis [631]. All published data support this notion, and several pathways have been identified that are involved in SCs’ ability to modulate the tumor microenvironment and promote tumor growth, spread, and progression to metastatic disease.

First, SCs stimulate the proliferation, motility, and invasiveness of tumor cells. In addition, SCs support both epithelial–mesenchymal and mesenchymal–epithelial transitions of malignant cells [632,633]. For instance, in vitro and in vivo, SCs supported the proliferation and migration of colon cancer cells [36] by secreting NGF and activating the TrkA/ERK/ELK1/ZEB1 signaling pathway in tumor cells [634] or via the activated NF-κB/IL-8 axis [635]. SCs could stimulate pancreatic cancer growth and invasion by releasing MMP-2, tissue inhibitor of metalloproteinases-2, galectin-3 binding protein, cathepsin D, proteoglycan biglycan, plasminogen activator inhibitor-1, galectin-1, or IL-6 [636,637]. SCs could also support the invasion and migration of prostate and pancreatic malignant cells [638].

Similar data have been reported for the lung cancer models. Exosomal miRNA-21-5p from human SCs (i) increased the proliferation, motility, and invasiveness of several human non-small cell lung cancer (NSCLC) cell lines in vitro by targeting metalloprotease inhibitor RECK in tumor cells and (ii) augmented human lung cancer cell progression and lymph node metastasis in vivo [639]. Human SCs have also been reported to enhance the proliferation and dissemination of various human small cell lung cancer (SCLC) cell lines in vitro and in vivo [640]. Mouse model experiments revealed that SCs promoted lung cancer cell proliferation and motility via upregulating M2 macrophage polarization or through secretory CXCL5/CXCR2-mediated activation of the PI3K/AKT/GSK-3beta/Snail-Twist signaling pathway in malignant cells [633,641]. Notably, SCs also stimulate the epithelial–mesenchymal transition (EMT) of malignant cells, supporting their invasive potential and metastasis formation, as has been reported for lung, pancreatic, salivary adenoid cystic carcinoma, and cholangiocarcinoma [633,637,642,643,644]. Therefore, understanding the metastasis pathways in demyelinating diseases, commonly comorbid with primary cancers, should offer new approaches to recognize how damaged SCs may influence and slow metastasis formation [645].

Importantly, SCs can chemoattract cancerous cells and stimulate perineural invasion or perineural spread—malignant cell dissemination in and along nerve bundles [629,646,647,648]. Tumor-activated SCs may facilitate the spread of cancerous cells along nerves by forming tracks for migration within the nerve and reorganizing malignant cells into chains [649,650]. For instance, the neuropeptide neuromedin B, produced by cervical cancer, can reprogram SCs to initiate perineural invasion by secreting CCL2 and directing axon regeneration [651]. Cervical cancer cells also upregulated the dedifferentiation of SCs, which in turn promoted perineural invasion by producing the chemokine FGF-17 and by degrading ECM with cathepsin S and MMP-12 [652]. The perineural invasion in prostate cancer was also associated with SCs [653]. SC-produced NGF and TGF-β may be related to a more aggressive phenotype of cholangiocarcinoma [644,654]. The presence of SCs has been shown to increase the aggressiveness of oral cancer cells by stimulating their proliferation, extracellular matrix breakdown, and altering cellular metabolism [655].

Second, SCs contribute to proper vascular remodeling. New data revealed that GFAP^+^ cells, derived from nerve-associated tissue-resident SCs, surrounded blood vessels in the microenvironments of melanoma, prostate, and breast cancer, and that genetic depletion of resident SCs attenuated tumor development by regulating angiogenesis and immunosurveillance [656]. The pro-angiogenic effects of SCs may include the promotion of endothelial cell proliferation, migration, and tube formation [602,657]. They may be mediated by SC-derived exosomes containing integrin-β1 [561,658]. In addition, multi-omics and bioinformatics analysis of non-myelinating SCs in gastrointestinal tumors revealed their association with “Activating invasion and metastasis” and “Inducing angiogenesis” [292].

Third, SCs can regulate the activity, polarization, and function of effector and regulatory immune cells within the tumor microenvironment, thereby manipulating the immunosuppressive potential of the tumor milieu. Recent data suggest that intratumoral SCs can attract immature myeloid cells, conventional dendritic cells, macrophages, and T cells and polarize them into immunosuppressive MDSC, regulatory DC, M2 tumor-associated macrophages (TAMs) and regulatory and exhausted T cells [496,498,499,641,659]. While the interaction between tumor-associated SCs and NK cells has not yet been documented, SC-derived TGF-β, PgE2, IL-6, and other molecules are known to control the activation and function of NK cells, suggesting a likely crosstalk between SCs and NK cells in the tumor environment [660]. New bioinformatic data confirm the association of non-myelinating SCs with MDSC, monocytes, neutrophils, regulatory T cells, and NK cells in gastrointestinal tumors [292]. SCs can also interact with mast cells, contributing to the formation of the tumor immunoenvironment [661]. These data prove that SCs actively participate in the recruitment and alteration of immune cells at the tumor site, forming, shaping, and maintaining the immunosuppressive and tolerogenic tumor immunoenvironment that facilitates immune escape, protection, proliferation, and spreading of malignant cells [629,660].

Fourth, SCs promote the conversion of cancer-associated fibroblasts (CAFs) into inflammatory CAFs, characterized by a more malignant phenotype, which accelerates tumor progression [662]. SC-derived pleiotrophin could upregulate CAF proliferation and collagen synthesis [663]. Ephrin, neuregulin-1 (NRG1), and tenascin-C signaling may play a role in SC and fibroblast communication [664,665,666].

Thus, various recent studies have revealed an active and essential role of SCs in promoting carcinogenesis and tumor growth [667,668,669]. Accumulating experimental data also suggest that the tumor-associated SCs in the tumor microenvironment play foundational roles in the metastatic transformation and metastatic dissemination of cancerous cells [645]. This implies that manipulation of SCs could supply a helpful approach to improve outcomes for patients with cancer [656].

### 9.2. Schwann Cell Alterations and Reprogramming in Cancer

SCs have been identified and characterized within the tumor microenvironment in multiple animal and human samples. These tumor-associated or tumor-reactive SCs demonstrated signs of a dynamic process of activation, proliferation, denervation, morphological flexibility, adaptive cellular reprogramming, increased motility, EMT, and stemness. Reprogramming and adopting a dedifferentiated phenotype make the functional state of tumor-activated SCs similar to that of repair SCs seen during nerve trauma and wound repair processes. Therefore, they have been termed ‘repair-like’ SCs [628,640,659,667,670,671]. However, the ‘injury response phenotype’ may be a more appropriate term (Figure 1).

The retrospective in silico analysis of tumor biopsies revealed that high expression of genes associated with SCs correlated with worse prognosis in melanoma patients [656]. The clinicopathological analysis similarly demonstrated that SCs were enriched in human colon cancer tissues and were associated with tumor metastasis and poor prognosis [36,634]. The evaluation of histological sections of pancreatic ductal adenocarcinoma also showed enrichment of SCs compared to non-tumor tissues, which was associated with a worse prognosis and could serve as an independent prognostic factor for overall survival [637,662]. A similar correlation between SC density and patient survival has been reported for lung cancer [672] and cholangiocarcinoma [644]. Importantly, the abundance of tumor-associated SCs in pancreatic ductal adenocarcinoma correlated with tumor-promoting immune cells’ infiltration and unfavorable patient outcomes [673].

A multi-omics evaluation of tumor-associated adaptively reprogrammed SCs confirmed the expression of various transformation markers associated with activation, dedifferentiation, proliferation, and denervation. Altered expression or secretion of cytokines, chemokines, growth factors, and other molecules by “repair-like” or “injury-responding” SCs has also been repeatedly demonstrated. Otani et al. found that tumor-activated SCs could express catecholamine-synthesizing enzymes, produce adrenaline, and establish an adrenergic microenvironment, which potentiates the chemoresistance of lung cancer cells [674]. Upregulation of chemoresistance in pancreatic adenocarcinoma cells by SC-conditioned medium has also been reported [643]. Melanoma-reprogrammed SCs exhibited dysregulated lipid oxidation, characterized by increased production of prostaglandins and lipoxins, which suppressed the activation of antitumor T cells [659]. Lung cancer-reprogrammed SCs can be characterized by the differentially expressed genes and microRNA [675]. Colorectal carcinoma factors upregulate the expression of IL-8 in SCs, which in turn upregulates tumor cell motility [635]. Pancreatic cancer cell-derived exosomes can upregulate the motility of SCs and thus facilitate perineural invasion [676]. Furthermore, the crosstalk between SCs and TAM is associated with increased perineural invasion and poor survival rates in these patients. TAM could activate SCs via the FGF/PI3K/Akt/c-myc/GFAP pathway, while SC-derived IL-33 could drive macrophages into the perineural milieu and assist in their M2 polarization [677]. Analysis of injury-induced neuroma confirms the presence of SCs in the repair status and significant infiltration by several subsets of macrophages [678].

Recent studies have also revealed the regulatory effects of glycometabolism reprogramming, which may control various intracellular signaling pathways at both the transcriptional and post-transcriptional levels, of the inflammatory responses within glial cells in the tumor milieu [679]. For instance, increased dietary intake of palmitic acid supports a pro-metastatic state in malignant cells, associated with SC activation and elevated tumor innervation [680]. Interestingly, one study suggested that SCs in premalignant lesions undergo adaptive reprogramming during early pancreatic cancer development, potentially encouraging a protective antitumor neuronal response [681].

Overall, a better understanding and streamlining of SC reprogramming pathways in the tumor milieu provide new targets and tools for destabilizing and destroying the tumor microenvironment as an additional means of a combinational cancer therapy approach.

### 9.3. Schwann Cells in Peripheral Neuropathies in Cancer Patients

Peripheral neuropathies are prevalent in patients with cancer and are a leading cause of cancer-related pain. Twenty percent of cancer pain is likely purely neuropathic in origin [682]. Cancer-associated neuropathies can be a direct or indirect complication of cancer or cancer treatment. The direct complications are due to nerve compression or infiltration by cancerous cells. The indirect cancer consequences may be the result of nutritional and metabolic deficits, coagulopathy, or infections, or, most often, they are due to neurotoxic chemotherapeutic agents—chemotherapy-induced peripheral neuropathies. Iatrogenic nerve injuries include radiotherapy-induced nerve damage or post-surgical complications. Less frequently, neoplastic neuropathies appear as a paraneoplastic neurological syndrome with immune-mediated manifestation. Paraneoplastic neuropathies may occur during the tumor’s growth and progression but are more likely seen as the tumor’s first manifestation associated with its identification [683].

#### 9.3.1. Schwann Cells in Chemotherapy-Induced Peripheral Neuropathy

Chemotherapy-induced peripheral neuropathy is a common dose-limiting side effect seen in cancer patients treated with neurotoxic chemotherapy [684]. It is often sensory-predominant with pain syndrome. For instance, PNS lesions are a common complication of breast cancer treatment, with up to 80% of survivors experiencing signs of PNS damage [685,686]. Less frequently, cancer patients develop paraneoplastic, immune-mediated, or neoplastic neuropathies. Chemotherapeutic drugs, such as anti-microtubule agents (taxanes), vinca alkaloids, platinum compounds, proteasome inhibitors (bortezomib), and thalidomide, are neurotoxic and have a high likelihood of inducing sensory or sensorimotor neuropathy or peripheral demyelination. Interestingly, patients with Charcot–Marie–Tooth disease can have increased susceptibility and severity of chemotherapy-induced peripheral neuropathy [687,688].

Experimental clinical data demonstrate that chemotherapy-induced peripheral neuropathy, associated with sensory and motor nerve abnormalities, ref. [689] may be at least partly mediated by the impairment of SCs. Notably, axonal degeneration (i.e., Wallerian degeneration) plays a major role in the pathogenesis of many neurodegenerative disorders, including chemotherapy-induced peripheral neuropathy [690]. However, there is a lack of evidence to show the role of SC transplantation in chemotherapy-induced neurotoxicity. Related studies have focused on the damage caused by chemotherapeutic agents to SCs and on stem cell therapy for chemotherapy-induced peripheral neuropathy [235].

Bortezomib-induced peripheral neuropathy is associated with remarkable SC demyelination in sciatic nerves resulting from impaired lysosomal function and induction of autophagy in SCs [691]. In vitro, bortezomib can augment the dedifferentiation of cultured SCs [692]. Mechanistical data showed that bortezomib-induced endoplasmic reticulum (ER) damage to SCs was accompanied by the downregulation of myelin gene expression and the expression of macrophage chemoattractants, which together suggest that adaptive responses of SCs to bortezomib-induced ER stress may participate in the development of bortezomib-induced peripheral neuropathy [693]. Paclitaxel-induced peripheral neuropathy in vivo stimulated the expression of activating transcription factor 3 (ATF3; a marker of cellular injury) in myelinating SCs in the sciatic nerve [694]. Cisplatin and oxaliplatin induce disruption of myelin formation and mitochondrial dysfunction in SCs, while paclitaxel induces SC dedifferentiation characterized by increased expression of p75 and galectin-3, and reduced expression of MBP [695]. Similarly, cisplatin and carboplatin in vitro triggered mitochondrial dysfunction and dedifferentiation in SCs [696]. Epirubicin-docetaxel therapy may also induce SC dedifferentiation and macrophage infiltration in sciatic nerves [697]. These and similar results suggest that the cytotoxic effects of chemotherapy on SCs may be important in the development of neurotoxic neuropathy in conjunction with their immediate damage to peripheral neurons. Thus, agents that inhibit drug-induced SC damage are expected to have therapeutic potential for this common pathology [698].

Although many strategies have been established, no specific intervention is presently endorsed for the prevention or relief of chemotherapy-induced peripheral neuropathy. Koyanagi et al. have recently demonstrated for the first time that SC-derived galectin-3, the β-galactoside binding protein serving as an immune cell chemoattractant, may play an important pro-nociceptive role in the development of taxanes (paclitaxel and docetaxel)-induced peripheral neuropathy via macrophage infiltration into the peripheral nerves [698]. The absence or pharmacologic inhibition of galectin-3 in experimental animals prevented paclitaxel-induced macrophage infiltration and mechanical hypersensitivity. Thus, therapies targeting SC-derived galectin-3 might represent a unique tactic to suppress taxane-related chemotherapy-induced peripheral neuropathy.

Remarkably, vitamin C was able to rescue SCs treated with bortezomib by inhibiting the cytotoxicity in SCs [699]. Vitamin C is critical for the SC myelination process, as it has been shown to upregulate 10 pro-myelinating genes in SCs. Dietary vitamin C deficiency may trigger peripheral nerve hypomyelination [700]. Recent screening for SC-targeting agents as candidates for chemotherapy-induced neuropathy treatment revealed that a selective phosphodiesterase PDE3 inhibitor, cilostazol, blocked paclitaxel-induced dedifferentiation of cultured SCs via cAMP/EPAC signaling and reduced sciatic nerve SC dedifferentiation in a mouse model of paclitaxel-induced peripheral neuropathy [701]. Moreover, a new study reveals that dihydromyricetin mitigates peripheral neuropathy and the toxic effects on SCs induced by Bortezomib, a proteasome inhibitor used in the treatment of multiple myeloma. Dihydromyricetin, a flavanonol extracted from the Japanese raisin tree, restores lysosome–autophagy activity in Bortezomib-affected SCs through the ERK/TFEB pathway, thereby significantly improving mechanical allodynia, sciatic nerve conduction, and demyelination associated with the adverse reaction of Bortezomib in the peripheral nervous system [702].

Although no data support the notion that SC transplantation might be a potential strategy for treating chemotherapy-induced peripheral neuropathy, new data suggest that mesenchymal stem/stromal cell therapy has a beneficial effect on improving symptoms [235,703]. It would be essential to determine whether MSCs can differentiate into SC-like cells after chemotherapy in vivo.

#### 9.3.2. Paraneoplastic Neuropathies

Paraneoplastic neuropathies are autoimmune and reflect the development of immune reactions to ectopically expressed antigens on neoplastic cells that mimic onconeural (intracellular) and neuronal surface (synaptic) molecules expressed by neurons and glial cells [704]. Among peripheral paraneoplastic neurological syndrome cases, sensory neuronopathy is the most frequent pathology. In contrast, other types of neuropathies, such as sensory and sensorimotor polyneuropathy, demyelinating neuropathies, and autonomic neuropathies, are less common [705]. Many onconeural antibodies have been well characterized, with anti-Hu and anti-Yo antibodies being the most common. While most identified onconeural antibodies are associated with CNS diseases, anti-Hu antibodies are linked with pure sensory neuronopathy [334]. Notably, almost 30% of patients with presumed paraneoplastic neuropathies do not demonstrate autoantibodies, as their antibodies have not yet been identified [704]. For example, analysis of autoantibodies in a cohort of patients with paraneoplastic neuropathies revealed that a significantly higher proportion of patients with sensory–motor neuropathy demonstrated serum IgM antibodies against SCs compared to healthy controls [706]. In a case report of a patient with paraneoplastic cerebellar degeneration associated with para-ovarian adenocarcinoma, circulating antibodies recognizing SCs have also been demonstrated [707].

### 9.4. Targeting Schwann Cells in Cancer

Tumor-induced activation of SCs may serve as a new target to weaken the tumor-protecting microenvironment. While surgical or pharmacological denervation of the local or systemic tumor milieu cannot be considered a feasible treatment approach to target tumor-associated neurons and SCs, at least today, focusing on activated SCs in cancer might provide a new reasonable direction. Although no clinical data directly verify that SC-based therapy is a realistic cancer treatment, we predict a thriving advancement of this approach. Recent results have shown that neuroblastoma cells respond to the primary repair SC secretome by enhancing neuronal differentiation and reducing proliferation [708]. Although cancer-specific, this proof-of-principle study supports the concept of selectively targeting intratumoral SCs as a promising approach for therapeutic intervention. This concept capitalizes on the unique liabilities of tumor-associated SCs, aiming to interrupt their supportive role in tumor progression and possibly improve the efficacy of standard cancer treatments.

Promising therapeutic strategies include the hampering of SC reprogramming, depletion of functionally blocked SCs, and disturbance of SC crosstalk with tumor-infiltrating immune cells [660]. As an example, noncytolytic lymphocytic choriomeningitis virus has been shown to target SCs preferentially, but not neurons, via alpha-dystroglycan expressed on SCs, rendering them defective or incapable of forming compact myelin sheaths [385]. This infection did not induce SC apoptosis or cytotoxicity but disrupted the assembly of SC-axon units. While it is unknown whether the localized infection of tumor-associated SCs may block their protumorigenic activity, these results open an opportunity to control SC function in the cancer milieu.

Depleting tumor-activated SCs with a small-molecule compound may affect the immune resistance of malignant cells, dramatically altering the immunosuppressive tumor microenvironment. In the in vivo pancreatic ductal adenocarcinoma model, SC reduction markedly improved the efficacy of immune checkpoint inhibitors and blocked tumor growth [673]. Genetic ablation of SCs in the melanoma model downregulated tumor growth and angiogenesis, which were associated with increased tumor infiltration by cytotoxic T cells, decreased infiltration by immunosuppressive regulatory cells, and thus upregulated immune surveillance [656].

Targeting tumor-associated SCs to prevent or suppress SC-composed perineural invasion is another promising clinical implication. The preclinical data demonstrated that a single fraction of low-dose radiation therapy may induce SC death and impair perineural invasion of non-irradiated malignant cells in both in vitro and in vivo nerve perineural invasion models [709]. Inhibition of perineural invasion in pancreatic cancer in vivo can also be achieved by controlling c-Jun activity with a c-Jun inhibitor in tumor-activated SCs [650]. Furthermore, knowing that tumor-derived NGF may activate SC autophagy, which in turn promoted perineural invasion in pancreatic cancer, Zhang et al. reported that targeting NGF and inhibiting autophagy blocked SC autophagic flux, stimulated SC apoptosis, and decreased the invasion of malignant cells to nerve fibers, showing marked therapeutic effects in the perineural invasion in a pancreatic cancer model [710].

Collectively, these studies support the notion that situating intratumoral SCs as promising therapeutic targets and manipulating SCs, as well as revealing the key mechanisms of the SC–cancer relationship, may provide a practical approach to improving cancer management and outcomes for cancer patients [711].

## 10. Tumorigenic Schwann Cells

Different studies revealed that overexpression of growth factor ligands or receptors, including EGFR, PDGF, NRG, SCF, HGF, and TGF-β1, in Schwann lineage cells, especially in combination with other gene mutations, like *NF1*, could significantly upregulate the formation of SC tumors [712,713]. For example, transgenic mice that lack both the *PTEN* and neurofibromatosis 1 (*NF1*) genes in SCs and SC precursor cells have demonstrated an augmented development of neurofibroma and high-grade peripheral nerve sheath tumors [714]. Transgenic mice overexpressing the growth factor NRG1 in SCs developed malignant peripheral nerve sheath tumors [715]. Transgenic mice heterozygous for a Trp53-null allele and overexpressing EGFR in SCs showed a substantial rise in neurofibroma and high-grade peripheral nerve sheath tumor formation. Similarly, modulation of EGFR and TP53 expression in immortalized human SCs markedly enhanced proliferation and anchorage-independent growth in vitro [713].

There are three main types of nerve sheath tumors: schwannoma, neurofibroma, and perineurioma. Schwannomas are exclusively composed of SCs, while neurofibromas comprise neoplastic SCs or Schwann-like perineural cells and other cells, including fibroblasts, neurons, and immune and endothelial cells [716]. Therefore, all schwannomas are densely positive for the S-100 protein, while neurofibromas may have more inconsistent S-100 expression.

Perineuriomas, a rare entity of peripheral nerve sheath tumors, in contrast to schwannomas or neurofibromas, are stained positive for epithelial membrane antigen (MUC1/CD227), but not for S100 protein; positive staining for neuron filament protein demonstrates axon fibers surrounded by layers of perineural tumor cells [717]. Perineurioma is a benign, slow-growing peripheral nerve sheath tumor that typically occurs in adolescence or young adulthood [718]. Although mutations in the *TRAF7* gene and the *NF2* gene have been associated with some perineurioma cases, a better understanding of the genetic signature of these relatively rare lesions should lead to targeted therapy and improved patient care [719].

Of note, a recent finding that chromaffin cells of the adrenal medulla are formed from SC precursors [720] may improve our understanding of the origin of neuroblastoma and pheochromocytoma, which most commonly arise from the adrenal gland region. Furthermore, finding that neuroblastoma cells, derived from aggressive tumors, respond to repair SC secreted factors with upregulated neuronal differentiation and decreased proliferation [708] might help identify promising candidates for the therapy of aggressive neuroblastoma.

### 10.1. Neurofibromatosis

Neurofibromatosis type 1, also known as von Recklinghausen syndrome, is a tumor predisposition genetic syndrome associated with a mutated copy of the *NF1* tumor suppressor gene on chromosome 17q11.2, which encodes neurofibromin, controlling cell growth and proliferation via the RAS signaling pathway [721]. Neurofibromas, especially cutaneous neurofibromas, are the most common tumors in patients with neurofibromatosis type 1, although they can also develop malignant peripheral nerve sheath tumors and gliomas [722]. The hallmark lesion is the neurofibroma, a benign peripheral nerve sheath tumor, and malignant tumors of the CNS and PNS [723,724]. Mutations in *NF1* in SCs are the initial tumor-forming event that leads to either complete loss or substantial reduction of neurofibromin function, resulting in hyperactivation of RAS and multiple downstream signaling pathways [712,725]. The key role of SCs in the development of neurofibromas was also demonstrated by showing that homozygous loss of NF1 in SCs is sufficient to induce tumors in different mouse models [726,727]. Ex vivo experiments also indicated that neurofibromin-deficient SCs had a significant growth advantage [728]. Furthermore, *NF1−/−* SCs exhibit superior pro-inflammatory transcriptional programs, producing various inflammatory cytokines and paracrine factors that mediate immune cell recruitment to the neurofibroma site [729]. Interestingly, analysis of Nf1-deficient SCs demonstrated that although neurofibromas could originate from adult SCs, the nerve environment could switch from tumor-suppressive to tumor-promoting at a site of injury [730].

Lesions of SC origin may pose a risk for malignant transformation into malignant peripheral nerve sheath tumors, which are aggressive and invasive soft tissue sarcomas with a poor prognosis derived from neurofibromatous SCs [731]. TAZ/YAP hyperactivation caused by Lats1/2-deficiency may be an oncogenic signaling hub that reprograms SCs to a cancerous, progenitor-like phenotype and promotes hyperproliferation, leading to malignant peripheral nerve sheath tumors [732]. Proteomic analysis of pathogenic SCs from malignant peripheral nerve sheath tumors identified over 17,000 distinctive peptides corresponding to more than 1500 individual proteins [733]. DAVID Genetic Association Database analysis, which explores the associations between the identified proteins and diseases, revealed that many SC-derived proteins were associated with cancer, neurological disorders, and immunological and infectious diseases.

Although treatment options, in addition to surgery, for neurofibromas and peripheral nerve sheath tumors are limited, various promising candidate therapeutic targets of signaling pathways for neurofibromatosis type 1-related malignant peripheral nerve sheath tumors have emerged in recent years [734,735]. However, there is an important concept of using cautiously engineered isogenic human Schwann lineage cell lines harboring neurofibromatosis type 1 applicable mutations to search for mutation-definite medication sensitivities and artificial lethal genetic interactions to determine novel therapeutic approaches [736,737].

The intricate interactions between tumor-origin SCs and neurofibroma-associated fibroblasts characterize neurofibromatosis type 1-associated plexiform neurofibroma. A recent study found that mutant SC-derived pleiotrophin, also known as heparin-binding brain mitogen, may be responsible for excessive collagen deposition by fibroblasts, a pathological hallmark of plexiform neurofibroma [663]. Inhibition of nucleolin, a pleiotrophin receptor, signaling reversed collagen synthesis in neurofibroma-associated fibroblasts, suggesting that targeting this pathway offers a new therapeutic strategy for plexiform neurofibroma. Interestingly, SCs deficient in neurofibromin, encoded by the *NF1* gene, are biomechanically incompatible, as they lack sensitivity to the biomechanical microenvironment, which likely plays a role in tumor initiation and progression [738]. NF1 is also required to conserve mitochondrial respiratory function in SCs through the stabilization of NADH-associated oxidative phosphorylation and electron transfer [739], confirming that it may serve as a promising drug target.

Neurofibromatosis 2 is less common than neurofibromatosis type 1 and is triggered by mutations of the *NF2* gene in chromosome 22, which encodes the tumor suppressor, merlin, and closely related cytoskeletal ERM proteins (ezrin, radixin, moesin) [722]. Mice expressing a mutated NF2 protein showed a high prevalence of SC-derived tumors and SC hyperplasia [740]. Vestibular schwannomas are the hallmark of neurofibromatosis 2, though non-vestibular schwannomas of the cranial, spinal, and peripheral nerves are also common, as are meningiomas.

New important data demonstrated that *NF2* inactivation drove the activation of p21-activated kinases (PAKs) signaling, which initiated NF1-mutant SC tumor dedifferentiation and resistance to therapy [741]. A comprehensive analysis of clinical human and experimental animal data suggest that NF1 or NF2 loss in SCs is necessary but not sufficient for transformation because other genetic harms and non-cellular independent factors play a crucial role in tumorigenesis [742]. A better understanding of the mechanisms triggering transformation and metastasis in SC lineages will eventually improve outcomes for patients who develop SC tumors.

### 10.2. Schwannomas

Schwannomas are the usual sporadic nervous system tumors and hallmarks of a tumor predisposition syndrome—neurofibromatosis type 2. Schwannomas (or neurilemmomas) are generally benign SC tumors that grow without the entrapment of axons, in contrast to other SC-derived tumors, like neurofibromas, which involve multiple fascicles of a nerve [743]. Less than 1% of schwannomas may become malignant, degenerating into neurofibrosarcoma. Schwannomas are caused by mutations in the loss-of-function NF2 tumor suppressor gene or a loss of chromosome 22, where the NF2 gene is located. The *NF2* gene encodes merlin (schwannomin). Merlin is involved in various signaling pathways, including PI3K/Akt/mTORC1, Ras/Raf/MEK/ERK, receptor tyrosine kinase, and Hippo signaling, to regulate cell growth, proliferation, and survival. Mutations in the *NF2* gene result in the production of a nonfunctional merlin protein, leading to uncontrolled cell proliferation and tumorigenesis [744].

Furthermore, in schwannomas, the longitudinal cues that naturally coordinate intrinsically polarized signaling in SCs are absent, and cells exhibit extremely varied polarized surface content. Recent data show that NF2-deficient SCs with unstable polarity can adopt different phenotypic statuses exhibiting enhanced multipolarity and exaggerated cytoskeletal responses to Nrg1 availability [745]. Differential signaling triggered by the two polarity states could also explain the heterogeneous therapeutic responsiveness observed for mTORC1- and EGFR-inhibiting drugs in schwannoma patients [746].

Schwannomas can develop in the cranial nerves or myelinated peripheral nerves, with nearly 25% of these tumors originating in the head and neck nerve structures. Among the cranial nerves, the vestibular, trigeminal, and hypoglossal nerves are most often affected by schwannomas [747].

#### 10.2.1. Schwannomas of Cranial Nerves and Spinal Cord

Schwannomas originating from SCs of cranial or spinal nerves are benign tumors [748]. The most common cranial schwannomas developing on the eighth cranial nerve are vestibular schwannomas. Schwannomas of other cranial nerves, such as the trigeminal, facial, and lower cranial nerves, are significantly less common.

A vestibular schwannoma, or acoustic neuroma, according to Rudolf Virchow, is a benign tumor that arises from SCs lining the vestibulocochlear nerve. These schwannomas may occur anywhere along the axons of the nerve from the glial–Schwann sheath junction [749]. Histologically, abnormal SCs exhibit several characteristic microscopic features, including biphasic architecture, nuclear palisading, a fibrous capsule, and degenerative signs [750]. Although the tumoral expansion of SCs in vestibular schwannoma may be one of the causes of affected hearing, demyelination of SCs, as the main morphologic damage of the auditory nerve, and the regulation of long-term survival and function of the auditory nerve, may be directly involved in hearing loss [751].

ScRNA-seq and exome sequencing data identified repair-like and MHC-II antigen-presenting SCs associated with myeloid cell infiltrate in the tumor microenvironment, implicating a nerve injury-like process. Injury-like SCs support tumor growth by attracting myeloid cells through CSF1 signaling pathways, which may be therapeutically targeted [752]. Of note, data showing that CSF1-R and c-Kit receptor inhibitor masitinib could prevent SC reactivity in degenerating nerves and ameliorate sciatic nerve pathology in ALS rats [196] provide an opportunity for pharmacological treatment of vestibular schwannomas. Furthermore, vestibular schwannomas that express markers of immature and denervated SCs also express NRG1 and activated ErbB2. Blocking anti-NRG1 and anti-ErbB2 inhibitory antibodies could inhibit SC proliferation, suggesting an autocrine pathway that stimulates tumor growth [753]. Thus, the inhibition of constitutive NRG/ErbB signaling provides therapeutic potential for patients with vestibular schwannomas. Similar pathological mechanisms may also be targeted in neurofibromatosis type 1 SC neoplasms known as malignant peripheral nerve sheath tumors [754,755].

Interestingly, recent data have demonstrated that losartan, an FDA-approved antihypertensive drug that blocks fibrotic and inflammatory signaling, can improve hearing loss in patients with neurofibromatosis type 2 by reducing fibrosis [756]. Although these experiments utilized mouse models, analysis of patient samples and data confirmed that losartan treatment normalized the schwannoma microenvironment, suggesting that these findings provide a rationale for prospective clinical trials [756].

Spinal schwannomas can be located intradurally, extradurally, or rarely intramedullary. Schwannomas can grow extramedullary as spinal cord tumors, representing one of the most common primary intradural extramedullary neoplasms, with malignant peripheral nerve sheath tumors being a less common entity [757]. Spinal schwannomas are among the most common intradural spinal cord tumors in adults, accounting for about 30–50% of such lesions [758]. They typically arise from spinal nerve roots and are associated with neurofibromatosis [759]. Extradural schwannomas can be distinguished from other nerve sheath tumors growing within the spinal canal by their clinicopathological features and their unlikely origin from the nerve root [760].

#### 10.2.2. Peripheral Schwannomas

Non-cranial nerve schwannomas are among the most common peripheral nerve sheath tumors in adults and can occur at a variety of anatomical locations. Up to 50% of such tumors appear in the head and neck. They are less common than schwannomas in cranial nerves and may present with unique symptoms. Peripheral schwannomas are benign, isolated, non-invasive, and encapsulated tumors that originate from differentiated SCs. Immunohistochemically, all peripheral schwannomas express S100 protein [761]. Almost 90% of peripheral schwannomas arise sporadically and are associated with the dysregulation of the tumor suppressor protein Merlin. However, familial tumor syndromes, like neurofibromatosis type 2 and schwannomatosis, can also cause peripheral schwannomas in 10% of cases [762].

Malignant transformation of benign schwannoma is rare; however, malignant alternates of schwannomas occur and account for approximately 5–10% of all soft tissue sarcomas [763]. Malignant peripheral schwannomas are more common than their cranial equivalents: they have been described in all somatic nerves of the PNS, including both peripheral and spinal nerves [764].

### 10.3. Schwannomatosis

Schwannomatosis is a rare inherited disorder characterized by the predisposition for developing multiple intracranial, spinal, or peripheral nerve schwannomas, with the most common symptom being chronic debilitating pain and neurological dysfunction. Affected individuals inevitably develop schwannomas, typically affecting vestibular nerves [765]. Schwannomatosis is commonly regarded as the least common form of neurofibromatosis [764]. Although historically, schwannomatosis is usually considered a form of neurofibromatosis type 2 [766], recently developed molecular and clinical diagnostic criteria suggest that these two conditions are clinically and genetically distinct entities, although they share many common features [767]. Neurofibromatosis type 2 and schwannomatosis may share the same trigger—loss-of-function mutations in the *NF2* gene. However, schwannomatosis is often genetically heterogeneous, and the NF2 gene mutation is not always involved. Mutations of other tumor suppressor genes, such as *SMARCB1* or *LZTR1*, on chromosome 22 have been identified in familial schwannomatosis cases. Nevertheless, schwannomatosis patients might still be misdiagnosed with neurofibromatosis type 2 or other nerve tumor diseases [768].

Unfortunately, the results of chromosome analyses of SCs in schwannomatosis or schwannomatosis predisposition syndrome are very restricted, which limits the development of potential cell and gene therapy approaches for treating schwannomatosis. However, there are xenograft mouse models with human *NF2*-related schwannomatosis tumors in the sciatic nerve of nude mice. Direct intratumoral injection of viral vectors supplying a functional copy of the mutated or inactivated *NF2* gene to augment functional Merlin protein re-expression in NF2-deficient tumor cells under the control of the SC-specific promoter P0 resulted in both the death of tumor cells and a significant regression of well-established tumors [532]. However, models focusing on *SMARCB1* or *LZTR1* mutations and their preclinical evaluation have not yet been developed. Nevertheless, the recent development of immortalized SC lines from human schwannomatosis tumors that retain their essential genotype, phenotype, and cell growth patterns should stimulate future studies on the molecular aspects, signaling, and cellular pathways, as well as therapeutic interventions, in different variants of schwannomatosis [769].

### 10.4. Schwannosis

The proliferation of ectopic SCs within the CNS parenchyma, or schwannosis, is not associated with a neoplastic process; however, in early life, it may be coupled with human neurofibromatosis type 2. In addition, transplanted SCs and resident SCs after injury may form schwannosis, which is different from SC tumors such as schwannoma since they normally form myelin sheaths and axons. A few animal cases confirm this pathological proliferation of SCs, described as proliferating cells with solid intracytoplasmic immunoreactivity for SC cell markers, such as MPZ and periaxin, suggesting the formation of PNS myelin within the spinal cord [770].

Interestingly, schwannosis, as the aberrant proliferation of SCs and nerve fibers, was also described in both human and experimental spinal cord injuries [771,772]. Based on the expression of GFAP and chondroitin sulfate proteoglycan (CSPG), Bruce et al. reported evidence of schwannosis in almost 50% of patients with spinal cord injury who survived 24 h to 24 years after injury [773]. A case of schwannosis in the brainstem has also been reported [774].

A benign proliferation of SCs and partial myelination of CNS axons may follow different chronic stimuli, including traumatic, degenerative, malignant, and compression lesions. Schwannosis may also result from a developmental abnormality, such as ectopia, during ontogenesis [774,775,776,777].

### 10.5. Transmissible Schwann Cell Cancers

Devil facial tumor disease (DFTD), an aggressive clonally transmissible non-viral malignancy, is a contagious cancer representing a very rare pathogenic phenomenon of cancerous cells spreading between genetically distinct hosts. DFTD affects Tasmanian devils (*Sarcophilus harrisii*), carnivorous marsupials endemic to Tasmania, during social biting behaviors and persists in the population in two distinct lineages of cancer: Devil Facial Tumor 1 and 2 (DFT1, DFT2) [778]. The inherent plasticity of SCs helps explain how a common progenitor cell can give rise to apparent contagious cancers. Proteomic and gene expression profiles confirmed that DFT1 and DFT2 are derived from SCs at different stages of differentiation and emerge in different individuals [779,780]. DFT2 reveals a molecular signature of less-differentiated SCs that express immunological markers associated with nerve repair [778]. Though DFT1 cells primarily display a myelinating phenotype, and DFT2 cells exhibit deactivation of myelination pathways, both malignant SC populations show upregulation of pathways associated with SC injury responses [781].

DFT1 tumors express many proteins associated with SC differentiation and myelination, such as S-100, MBP, MPZ, nestin, PMP22, and the nerve growth factor receptor (NGFR) [782]. DFT2 cells express S-100, SOX10, nesting, and NGFR [783]. Analysis of Tasmanian devil normal PNS samples revealed the expression of a spectrum of myelin genes, suggesting that these functions are intact and SCs are conserved [781].

Cancer cells are transmitted as malignant allografts among Tasmanian devils, with no detectable induction of immune-mediated allograft rejection. The loss of MHC class 1 molecule expression on the surface of DFT1 cells, resulting from epigenetic downregulation of several components of the MHC class I antigen processing machinery, cannot explain the phenomenon of immune escape, as DFT2 cells express MHC class I molecules. New biochemical data revealed that MHC molecules on DFT2 cells are similar to those of the infected host devils, explaining how malignant SCs may overcome antitumor and allogeneic immunological barriers [780]. Further experiments demonstrated that cancerous DFT2 SCs might slowly evolve to lose their MHC molecules, becoming as contagious as DFT1 cells. New insights into regulating MHC-I and MHC-II antigen processing and presentation in malignant SCs that overcome allogeneic barriers should open up opportunities for treating DFTD, which has a nearly 100% fatality rate [784]. The development of several vaccination platforms is in progress now [785,786]

### 10.6. Other Schwann Cell Tumors

Mucosal SC hamartomas, introduced in 2009 and considered gastrointestinal schwannomas [787], are benign, intramucosal tumors of mesenchymal origin commonly located in the distal colon [788]. The term was used to distinguish them from true “neuromas” and “neurofibromas” [789]. SC hamartomas were also found in the gallbladder, the esophagogastric union, and the antrum, and they are generally asymptomatic. Mucosal Schwann cell hamartomas are primarily observed as small polyps, and diagnosis is made after excluding other lesions that resemble spindle cell proliferation and other neuronal tumors [790]. The identification relies on the distinctive clinicopathologic, histologic, and immunohistochemical features. These SC tumors are characteristically positive for S-100 protein, but CD117/c-Kit^neg^, CD227/MUC1^neg^, CD34^neg^, neurofilament protein^low/neg^, and negative for smooth muscle actin staining, which together distinguish mucosal SC hamartomas from other gastrointestinal neoplasms [791].

Because SC hamartomas are rare and benign, no preclinical models or clinical studies have focused on the pharmacological targeting of tumorigenic SCs. There are no data on isolation, culture, or phenotypic and functional characterization of SCs from these tumors.

## 11. Conclusions and Future Directions

SCs demonstrate incredible plasticity and a regulable and reversible adaptive cellular reprogramming response to differential stimuli. They are proven to play an etiological and/or pathogenic role in a spectrum of peripheral nerve pathological conditions and diseases. In inherited demyelinating neuropathy, traumatic peripheral neuropathies, immune-mediated neuropathies, and infection-induced demyelination of the PNS, as well as peripheral neuropathic pain, wound healing, and cancer development and progression, pathologically dedifferentiated, dysmyelinated, or repolarized SCs constitute the main nerve pathology. A significant number of evidence demonstrates the beneficial therapeutic roles of SCs and SC-like cells in peripheral neuropathy. Targeting pathological SCs, implicated in specific diseases, is a promising approach to treating peripheral neuropathies (Figure 1 and Figure 2). SC-based therapy also represents an attractive and efficient tactic for treating peripheral neuropathy (Figure 3). Although a growing body of evidence suggests that SCs are a promising target and encouraging tool for advanced experimental and clinical studies, several critical caveats and unresolved issues require attention.

Given their proven role in tissue repair, transplanted SCs are widely evaluated as a new tool for nerve repair in preclinical studies and clinical trials. However, SC transplantation is limited by the necessity of a nerve biopsy to isolate SCs and the lengthy cell culture time required to prepare a sufficient number of cells [256]. Additionally, autologous SC generation protocols are hindered by the limited proliferative activity of adult nerve-associated SCs in vitro. This might result in SC senescence ex vivo, seriously reducing the regenerative capacity [39,73,792]. Improved strategies are also needed to effectively obtain and/or produce functional repair SCs in a shorter timeframe. Understanding these requirements, nerve-derived SCs cannot be considered the optimal source for clinical purposes [793]. Hence, alternative sources of SCs are expected to facilitate the faster development of efficient glial support cell therapies (Figure 3) and are thus being actively explored [237].

One alternative approach is the in vitro differentiation of SCs from other cell types. Cell sources, differentiation techniques, functional cell characterization, and protocol efficiency have been described and compared [794]. Stem cells represent a valuable resource for SC generation [235]. Mesenchymal stem/stromal cells that can be instantly harvested from different tissues, like skin, bone marrow, umbilical cord, and adipose tissue, are an excellent source of SC-like cells [795]. For example, SCs prepared from bone marrow stromal cells were similar to sciatic nerve-derived SCs in their ability to support axonal regeneration in several animal models [235,794]. Similarly, SC-like cells prepared from subcutaneous adipose-derived stem cells demonstrated higher engraftment and migration capacity than naïve SCs after transplantation into the injured sciatic nerve [796]. SC precursors can be obtained from human pluripotent stem cells cultured and differentiated with TGF-β and GSK-3 inhibitors and neuregulin-1 NRG1 [797].

Easily accessible human skin fibroblasts may represent another unusual source of SCs, as recently reported using the direct conversion-reprogramming approach to prepare SC precursors proficient in differentiating into functional SCs [798]. Injection of these SCs enabled the repair of a crushed sciatic nerve in a mouse model. Direct conversion technology, which bypasses the pluripotent stage and irreversibly alters cell types, has demonstrated its utility for SC generation by modulating specific transcription factors involved in SC development [237,799].

The therapeutic utilization of SCs also requires an improved understanding of the mechanisms of SC-induced tissue and nerve regeneration and remyelination. The increasing evidence on the capacity of SCs to support wound healing and tissue regeneration is a thrilling field of future investigation. Available results suggest that some SC-mediated effects on regeneration are indirect, resulting from the release of factors and paracrine signaling mechanisms. It is crucial to determine whether SCs are the primary providers of these communications and whether these results can be succeeded through other direct means. Moreover, combining SCs or SC-derived factors with peripheral nerve repair biomaterials appears to be a promising addition to efficiently guide the regeneration of axons and facilitate the myelination of nerve fibers. The utilization of Artificial Intelligence technology, equipped with Machine Learning and Deep Learning, may help develop neural and engineered regenerative biomaterials, thereby increasing the potential to enhance the field of SC-based treatments and their clinical applicability.

Another essential consideration is identifying the SC subtypes that mediate the reported or expected therapeutic effects. The targeted or transplanted SCs can vary, depending on the type of tissues, injury, and repair pathway. The associated vector of future investigation is the detection of biomarkers to prospectively harvest SCs with a repair phenotype [237]. Our understanding of SC alterations with aging and how these changes contribute to their dysfunction, disease pathogenesis, and therapeutic use remains limited [800,801].

Understanding the specific signal transduction and metabolic pathways within normal and pathogenic SCs that are regulated by the mRNA–microRNA–lncRNA competitive endogenous RNA network is a promising clinical research direction. These studies should identify novel intracellular targets that enable the stabilization of functionally abnormal SCs and the generation of therapeutic SCs with a specific pattern of functions. Anticipated results may also help prepare therapeutic cell-free SC-derived exosomes with the required content and activity. By leveraging newly developed technologies, a quickly expanding toolbox for in vitro and ex vivo modeling, modern bioengineering-based approaches, and readily available bioinformatic and database resources, it is possible to create high-throughput, adaptive systems for characterizing and modulating SCs from individual patients. Altogether, this will lead to significant improvements in our understanding of neuropathy pathogenesis and treatments, as well as in personalized medicine, SC-targeted therapeutics, and healthcare overall.

## Figures and Tables

**Figure 1 cells-14-01336-f001:**
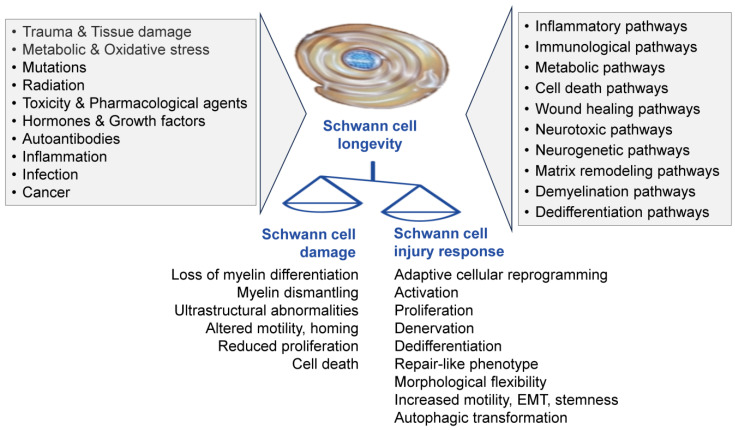
Alteration of Schwann cell longevity, phenotype, and function in different diseases. SCs are implicated in a wide range of diseases and pathological conditions due to their inherent potential for phenotypic and functional plasticity, as well as their ability to interact with various subsets of surrounding cells. Many inherited, toxic, autoimmune, metabolic, or pharmacological stimuli can damage SC motility, myelin production, and survival. In contrast, other stimuli, such as trauma, hormones, infection, or inflammation, induce SC injury response. Associated adaptive cellular reprogramming of SCs is seen as their denervation, dedifferentiation, proliferation, and phenotypic flexibility. Both SC damage and adaptive reprogramming are associated with significant and often specific alterations in cellular pathways, including neurotoxic, neurogenic, inflammatory, matrix remodeling, wound healing, and cell death pathways. Together with abundant experimental results, these data suggest that SCs can serve as therapeutic targets or cell transfer therapy for various types of neuropathies and associated pathophysiological conditions.

**Figure 2 cells-14-01336-f002:**
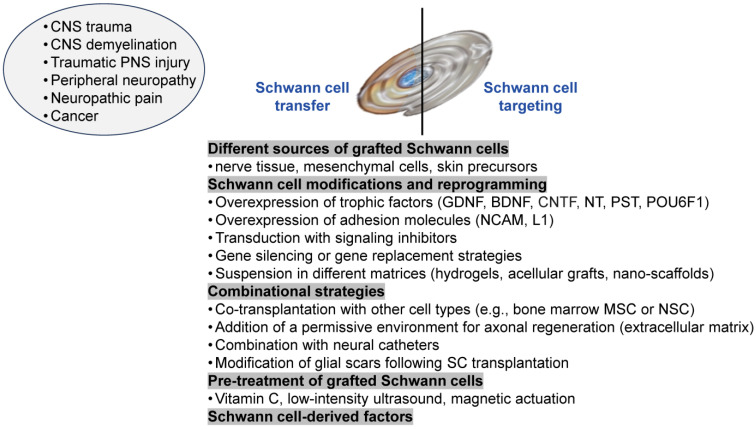
Employment of Schwann cell targeting and Schwann cell transplantation alone or in combination with known neuroprotective and neuroregenerative strategies for the treatment of CNS and PNS injury and neuropathy. Many pathological conditions, such as PNS and CNS trauma, injury, and demyelination, as well as neuropathies of different origins, chronic pain syndrome, and cancer, can benefit from targeting abnormal SCs or transplanting naïve or bioengineered SCs, as has been shown in numerous preclinical and clinical studies. The transfer of SCs alone or in combination with different scaffolds or neurotrophic factors significantly accelerates and improves the regeneration and myelination of injured spinal cord and peripheral nerves. Natural and pharmacological agents block high glucose-induced oxidative damage and apoptosis in SCs, enhance SC myelination, and thereby inhibit peripheral nerve degeneration, accelerating nerve conduction velocity. Many pharmacological agents, inhibitors of signal transduction pathways, and natural plant molecules can stimulate SC viability, proliferation, and migration and upregulate the expression of myelin-related and nerve growth-promoting genes in SCs, therefore promoting peripheral nerve regeneration. Gene therapy approaches, such as gene silencing or gene replacement strategies, targeting SCs demonstrate promise in treating inherited peripheral neuropathies, like Charcot–Marie–Tooth disease. SC-derived exosomal vesicles may have therapeutic value in repairing the wounded spinal cord and chemotherapy-induced peripheral neuropathy. SCs can also relieve neuropathic pain by remyelinating injured nerves. Although the role of SCs in the pathogenesis of various neuropathies is well established, and the applicability of SCs in treating these diseases has been proven, the full scope of potential SC targeting or the feasibility of their therapeutic application in CNS and PNS neuropathic conditions remains to be fully deciphered. BDNF, brain-derived neurotrophic factor; CNTF, ciliary neurotrophic factor; GDNF, glial cell line-derived neurotrophic factor; L1 (L1CAM), transmembrane protein member of the L1 protein family; MSC, mesenchymal stromal cell; NCAM, neural cell adhesion molecule; NT, neurotrophin; NSC, neural stem cell; POU6F1, POU domain class 6, transcription factor 1; PST, polysialyltransferase.

**Figure 3 cells-14-01336-f003:**
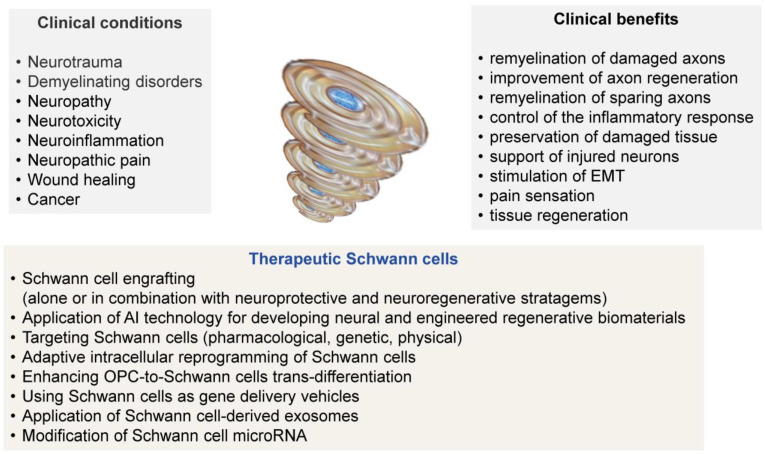
Basic strategies and clinical benefits of therapeutic Schwann cells. Numerous clinical and experimental studies have demonstrated the benefits of targeting pathogenic SCs and utilizing autologous or allogeneic naïve and engineered SCs, as well as their products, as a promising novel therapeutic approach. Proven clinical responses have been documented for various neuropathological conditions associated with neurotrauma, nerve injury, neurotoxic and neuroinflammatory disorders, neuropathic pain, demyelinating states, as well as wound healing and cancer-related abnormalities. The regenerative effect conferred by SCs is not limited to strong capabilities in accelerating axon repair, regeneration, and remyelination. Transplantation of reprogrammed or engineered SCs, application of SC exosomes, and pharmacological targeting or redifferentiation of resident SCs and their precursors in preclinical and clinical studies can improve the control of pain sensation, inflammatory response, EMT, and tissue regeneration, thereby preserving damaged tissue and supporting injured neurons. The utilization of Artificial Intelligence technology should accelerate the development of practical regenerative biomaterials, thereby increasing the potential of SC-based treatments and their clinical applicability. It remains to be established what the optimal conditions are for utilizing SC-based approaches and crucial signal transduction pathways in pathological and therapeutic SCs, which should be targeted for treating specific neurological diseases and abnormalities. AI, Artificial Intelligence; EMT, epithelial–mesenchymal transition; OPC, oligodendrocyte progenitor cells.

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
