# Peer review of "Pathologic and Therapeutic Schwann Cells"

_cells, 2025, doi:10.3390/cells14171336_

Round 1
Reviewer 1 Report
Comments and Suggestions for Authors
The review is well-written and provides a thorough overview of the diverse roles played by Schwann cells across various pathological contexts. I have no major concerns. My only suggestion is to consider including a few additional references.
In this sentence: This might be mediated by a compromised synthesis of the neuromuscular junction regulator agrin, or secretion of matrix metalloproteinase (MMP), or generation of CXCL12α (stromal cell-derived factor, SDF-1) in perisynaptic SCs, or SC-induced Connective Tissue Growth Factor-related pathways 186.
Instead of the review I would cite the original works. Here some suggestions:
Negro S., Lessi F., Duregotti E., Aretini P., La Ferla M., Franceschi S., Menicagli M., Bergamin E., Radice E., Thelen M., et al. CXCL12alpha/SDF-1 from perisynaptic Schwann cells promotes regeneration of injured motor axon terminals. EMBO Mol. Med. 2017;9:1000–1010. doi: 10.15252/emmm.201607257.
Chao T., Frump D., Lin M., Caiozzo V.J., Mozaffar T., Steward O., Gupta R. Matrix metalloproteinase 3 deletion preserves denervated motor endplates after traumatic nerve injury. Ann. Neurol. 2013;73:210–223. doi: 10.1002/ana.23781.
Negro S., Lauria F., Stazi M., Tebaldi T., D’Este G., Pirazzini M., Megighian A., Lessi F., Mazzanti C.M., Sales G., et al. Hydrogen peroxide induced by nerve injury promotes axon regeneration via connective tissue growth factor. Acta Neuropathol. Commun. 2022;10:189. doi: 10.1186/s40478-022-01495-5.
In the Neurofibromatosis paragraph (10.1.) I suggest the authors to include citations of important works about SCs and NF1 neurofibromatosis. Some suggestions:
Neurofibroma development in NF1 – insights into tumour initiation, Trends in Cell Biology, Volume 19, Issue 8, 2009,
Sara Ribeiro, Ilaria Napoli, Ian J. White, Simona Parrinello, Adrienne M. Flanagan, Ueli Suter, Luis F. Parada, Alison C. Lloyd, Injury Signals Cooperate with Nf1 Loss to Relieve the Tumor-Suppressive Environment of Adult Peripheral Nerve, Cell Reports, Volume 5.
Author Response
Comment 1:
In this sentence: This might be mediated by a compromised synthesis of the neuromuscular junction regulator agrin, or secretion of matrix metalloproteinase (MMP), or generation of CXCL12α (stromal cell-derived factor, SDF-1) in perisynaptic SCs, or SC-induced Connective Tissue Growth Factor-related pathways 186.
Instead of the review I would cite the original works.
Comment 2:
In the Neurofibromatosis paragraph (10.1.) I suggest the authors to include citations of important works about SCs and NF1 neurofibromatosis. Some suggestions: ...
Response:
We thank the Reviewer for the time and effort to read the manuscript. We are happy that there are no “major concerns”. We also thank the Reviewer for the suggestion to modify two paragraphs by citing original publications instead of reviews and providing these citations. Both parts have been revised in the updated manuscript.
Reviewer 2 Report
Comments and Suggestions for Authors
The paper provides a review of the role of Schwann cells (SCs) in both pathological conditions and therapeutic applications, with wide range of topics, from biology to innovative therapies. It is written more as a book chapter and needs to include literature search methodology and definition - narrative, scoping, comprehensive or systematic review, inclusion/exclusion criteria etc. The paper is quite lengthy and dense, keypoints are missing; the figures and tables could be more visually engaging or simplified; simplify technical terms
Author Response
We thank the Reviewer for reading the manuscript. We agree that the review “is quite lengthy and dense” and “is written more as a book chapter”. We want to stress that the manuscript was prepared for the Section titled “Cells of the Nervous System”, and thus represents a comprehensive review, as requested. In addition, the Special Issue of the Cells requested manuscripts covering “Emerging Roles of Glial Cells in Human Health and Disease”, suggesting that the review should be comprehensive and systematic. Therefore, our manuscript analyzes pre-clinical and clinical results of targeting cell signaling and genetic abnormalities in Schwann cells, as well as the outcomes of administering naïve and engineered Schwann cells, and Schwann cell-derived molecules/exosomes in the treatment of neuropathies of different origins. To the best of our knowledge, this is the first review on therapeutic Schwann cells in different neuropathies associated with traumatic, toxic, metabolic, infectious, inherited, and inflammatory conditions. All of these elements make the review unique, and we hope that the Reviewer agrees with us that the review will be both interesting and helpful to many readers.
Reviewer 3 Report
Comments and Suggestions for Authors
This is a well written comprehensive review of Schwann cells.
Abstract: Please define “PNS”.
Throughout the article, authors mention “oxidative stress”. Please describe more in detail the characteristics of “oxidative stress” for each situation.
In ALS, what does SOD1(G93A) mutant do to the activity of SOD1? Authors state “SCs expressing dismutase active mutant SOD1….” Does this mean that SOD1 activity is enhanced in this mutant? Please clarify.
Please describe the consequence of SOD enhancement in terms of hydrogen peroxide levels and also the function of peroxiredoxin, so that readers can better appreciate the implications of these findings related to ALS.
Author Response
Reviewer 3.
We thank the Reviewer for the time and effort to read the manuscript and for judging it as “a well-written comprehensive review of Schwann cells.”
Abstract: Please define “PNS”.
We are sorry for missing this. It was corrected.
Throughout the article, authors mention “oxidative stress”. Please describe more in detail the characteristics of “oxidative stress” for each situation.
We thank the Reviewer for this suggestion. Below are examples of the requested modifications to the corresponding parts of the revised manuscript.
2.2.2. Parkinson’s disease and Schwann cells
The oxidant-antioxidant theory, as one of several hypotheses proposed to explain the etiology of Parkinson’s disease, asserts that oxidative damage triggered by reactive oxygen species can promote dopaminergic neuron degeneration {Dorszewska, 2021 #1006}. Accumulating results revealed that oxidized lipids, proteins, and nucleic acids accumulate in the brain tissues of patients with Parkinson’s disease, suggesting that free radicals may affect both α-synuclein aggregation and disease progression {Caproni, 2025 #1007}. Although a few pathogenic pathways have been implicated in Parkinson’s disease, including mitochondrial dysfunction and oxidative stress, altered iron metabolism and ferroptosis, failure of the ubiquitin-proteasome system and impaired protein clearance, the processes of apoptosis and/or autophagy, and neuroinflammation, their interaction and cellular involvement are not yet entirely understood {Wal, 2022 #318}.
- Alzheimer's disease and Schwann cells
Alzheimer's disease is a neurodegenerative disorder manifested by extracellular amyloid beta plaques in the grey matter, intraneuronal hyperphosphorylated tau filaments, neuronal death, synapse elimination, and brain atrophy, and is associated with free radical and oxidative stress, metabolic dysregulation, and upregulated proinflammatory cytokines {Self, 2023 #339}. For instance, upregulated generation of reactive oxygen and reactive nitrogen species and defective mitochondrial dynamic balance can lead to the misfolding of amyloid-beta and hyperphosphorylation of tau proteins, affecting tau protein kinase activation and phosphatase inhibition, thus causing the formation of neurofibrillary tangles, a hallmark of Alzheimer's disease {Bhatia, 2021 #1008}. Dysfunctional mitochondria may further release ROS, increase the production of pro-apoptotic proteins, and cause apoptosis of neurons {Bai, 2022 #1009}.
3.2.1. Schwann cells in diabetic neuropathy
Advanced glycated end products, inflammation, and oxidative stress can damage SCs, decrease the production of NGF, and thus provoke axonal degeneration {Suzuki, 2004 #207;Sekido, 2004 #199;Naruse, 2019 #421;Mizisin, 2014 #429}. For instance, SCs, damaged by oxidative stress-induced mitochondrial dysfunction, demonstrated significantly decreased viability and increased apoptotic death associated with Bcl2, NF-κB, mTOR, Wnt signaling {Chao, 2017 #1010}{Tiong, 2019 #1011}.
In ALS, what does SOD1(G93A) mutant do to the activity of SOD1? Authors state “SCs expressing dismutase active mutant SOD1….” Does this mean that SOD1 activity is enhanced in this mutant? Please clarify.
We thank the Reviewer for this note. Functional activity of mutated SOD1 in ALS and SCs was clarified in the revised version of the manuscript, as shown below.
Cytoplasmic and mitochondrial SOD1 convert superoxide radicals into molecular oxygen and hydrogen peroxide. Mutations in SOD1, associated with alterations in SOD1 functionality and/or aberrant SOD1 aggregation, contribute to ALS pathogenesis. Rodent transgenic ALS models, developed by expressing a human SOD1 transgene with ALS-associated mutation G93A, reproduce the major phenotypic features of human ALS {Nagai, 2001 #1012}. Importantly, the development of motor neuron disease is not due to a loss of SOD1 function as many mutant forms of SOD1 retain nearly normal or even elevated SOD1 activity {Ripps, 1995 #1016}{Reaume, 1996 #1015}. Thus, while neither upregulated expression of wild-type SOD1 nor obliteration of endogenous SOD1 caused motor neuron disease, it is evident that disease is induced by an acquired toxicity of mutant SOD1 independent of its dismutase activity {Boillée, 2006 #1017}.
Increased SOD1 activity can ameliorate the production of ROS in the mitochondrial intermembrane space, contribute to mitochondrial damage in SOD1G93A rats {Ahtoniemi, 2008 #1013}. Therefore, mechanisms critical for ALS progression may include mitochondrial dysfunctions, excitotoxicity, oxidative stress, and changed Ca2+ metabolism. The effects of SOD1 G93A mutation in non-neuronal cells, such as glial cells, in ALS may be associated with the altered redox balance and perturbed expression of Ca2+ transporters that may be responsible for altered mitochondrial Ca2+ fluxes {Peggion, 2022 #1014}.
Please describe the consequence of SOD enhancement in terms of hydrogen peroxide levels and also the function of peroxiredoxin, so that readers can better appreciate the implications of these findings related to ALS.
We thank the Reviewer for this note. Functional activity of mutated SOD1 in SCs and the role of SC-derived peroxiredoxin were clarified in the revised version of the manuscript, as shown below.
SCs expressing a dismutase-active mutant SOD1 were shown to reduce disease progression in ALS mice, as the removal of mutant SOD1 reduced survival. This suggests a link between slow disease progression in ALS mice and a protective impact of dismutase active mutant SOD1 in SCs {Lobsiger, 2009 #546}. Importantly, SCs transfected with mutant SOD1 expressed low levels of peroxiredoxin 1, and the expression of peroxiredoxin 1 mRNA was significantly decreased in the lumbar spinal cord of SOD1G93A mice {Yamamuro-Tanabe, 2023 #550}. Peroxiredoxin 1, an antioxidant molecule derived from SCs, protected motor neurons from hydrogen peroxide-induced cell death, suggesting that the reduction of peroxiredoxin secreted from SCs contributes to increased ROS and accompanies motor neuronal death in ALS {Yamamuro-Tanabe, 2023 #550}. Similarly, peroxiredoxin 6 secreted by SCs has been shown to significantly inhibit neuron apoptosis and improve neurological recovery in different models {Tang, 2021 #1019}.
Round 2
Reviewer 3 Report
Comments and Suggestions for Authors.